# Fair Sparse Regression with Clustering: An Invex Relaxation for a Combinatorial Problem

**Adarsh Barik**
Department of Computer Science
Purdue University
West Lafayette, Indiana, USA
abarik@purdue.edu

Jean Honorio
Department of Computer Science
Purdue University
West Lafayette, Indiana, USA
jhonorio@purdue.edu

## Abstract

In this paper, we study the problem of fair sparse regression on a biased dataset where bias depends upon a hidden binary attribute. The presence of a hidden attribute adds an extra layer of complexity to the problem by combining sparse regression and clustering with unknown binary labels. The corresponding optimization problem is combinatorial, but we propose a novel relaxation of it as an *invex* optimization problem. To the best of our knowledge, this is the first invex relaxation for a combinatorial problem. We show that the inclusion of the debiasing/fairness constraint in our model has no adverse effect on the performance. Rather, it enables the recovery of the hidden attribute. The support of our recovered regression parameter vector matches exactly with the true parameter vector. Moreover, we simultaneously solve the clustering problem by recovering the exact value of the hidden attribute for each sample. Our method uses carefully constructed primal dual witnesses to provide theoretical guarantees for the combinatorial problem. To that end, we show that the sample complexity of our method is logarithmic in terms of the dimension of the regression parameter vector.

## 1 Introduction

In modern times, machine learning algorithms are used in a wide variety of applications, many of which are decision making processes such as hiring (Hoffman et al., 2018), predicting human behavior (Subrahmanian & Kumar, 2017), COMPAS (Correctional Offender Management Profiling for Alternative Sanctions) risk assessment (Brennan et al., 2009), among others. These decisions have large impacts on society (Kleinberg et al., 2018). Consequently, researchers have shown interest in developing methods that can mitigate unfair decisions and avoid bias amplification. Several fair algorithms have been proposed for machine learning problems such as regression (Agarwal, 2019; Berk, 2017; Calders, 2013), classification (Agarwal et al., 2018; Donini et al., 2018; Dwork et al., 2012; Feldman et al., 2015; Hardt et al., 2016; Huang & Vishnoi, 2019; Pedreshi et al., 2008; Zafar et al., 2019; Zemel et al., 2013) and clustering (Backurs et al., 2019; Bera et al., 2019; Chen et al., 2019; Chierichetti et al., 2017; Huang et al., 2019). A common thread in the above literature is that performance is only viewed in terms of risks, e.g., misclassification rate, false positive rate, false negative rate, mean squared error.

In the literature, fairness is discussed in the context of discrimination based on membership to a particular group (e.g. race, religion, gender) which is considered a sensitive attribute. Fairness is generally modeled explicitly by adding a fairness constraint or implicitly by incorporating it in the model itself. There have been several notions of fairness studied in linear regression. Berk (2017) proposed notions of individual fairness and group fairness, and modeled them as penalty functions. Calders (2013) proposed the fairness notions of equal means and balanced residuals by modeling them as explicit constraints. Agarwal (2019), Fitzsimons (2019) and Chzhen et al. (2020) studied

35th Conference on Neural Information Processing Systems (NeurIPS 2021).

Table 1: Comparison to prior work. Notation: $s$ is the number of non-zero entries in the regression parameter vector and $d$ is its dimension. The terms independent of $s$ and $d$ are not shown in the order notation.

| Paper | Hidden sensitive attribute | Modeling type | Sample complexity |
|---|---|---|---|
| Calders (2013); Agarwal (2019); Fitzsimons (2019) | No | Explicit constraint | Not provided |
| Berk (2017) | No | Penalty function | Not provided |
| Chzhen et al. (2020) | No | Implicit | Not provided |
| **Our paper** | **Yes** | **Implicit** | $\Omega(s^3 \log d)$ |

demographic parity. While Agarwal (2019), Fitzsimons (2019) modeled it as an explicit constraint, Chzhen et al. (2020) included it implicitly in their proposed model.

All the above work assume access to the sensitive attribute in the training samples and provide a framework which are inherently fair. Our work fundamentally differs from these work as we do not assume access to the sensitive attribute. Without knowing the sensitive attribute, it becomes difficult to ascertain bias, even for linear regression. In this work, we focus on identifying unfairly treated members/samples. This adds an extra layer of complexity to linear regression. We solve the linear regression problem while simultaneously solving a clustering problem where we identify two clusters – one which is positively biased and the other which is negatively biased. Table 1 shows a consolidated comparison of our work with the existing literature.

Once one identifies bias (positive or negative) for each sample, one could perform debiasing which would lead to the fairness notion of equal means (Calders, 2013) among the two groups (See Figure 1). It should be noted that identifying groups with positive or negative bias may not be same as identifying the sensitive attribute. The reason is that there may be multiple attributes that are highly correlated with the sensitive attribute. In such a situation, these correlated attributes can facilitate indirect discrimination even if the sensitive attribute is identified and removed. This is called the red-lining effect (Calders, 2010). Our model avoids this by directly identifying biased groups.

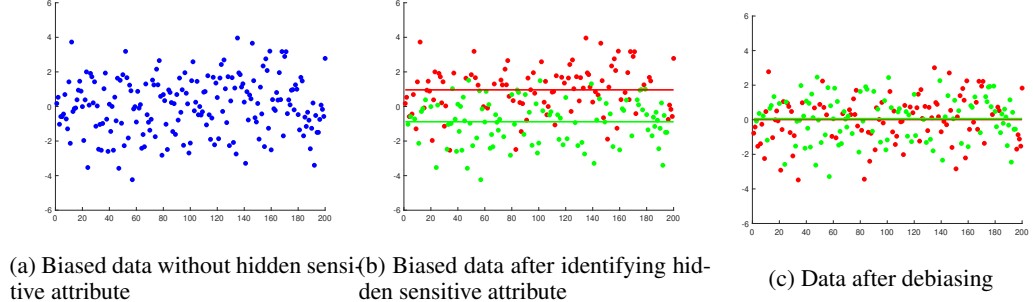

(a) Biased data without hidden sensitive attribute

(b) Biased data after identifying hidden sensitive attribute

(c) Data after debiasing

Figure 1: Data before debiasing and after debiasing. Notice how means for two groups (shown as horizontal lines) become almost equal after debiasing.

While the standard algorithms solving the sparse/LASSO problem in this setting do provide an estimate of the regression parameter vector, they do not fit the model accurately as they fail to consider any fairness criteria in their formulations. It is natural then to think about including the hidden attribute in LASSO itself. However, this breaks the convexity of the loss function which makes the problem intractable by the standard LASSO algorithms. The resulting problem is a combinatorial version of sparse linear regression with added clustering according to the hidden attribute. In this work, we propose a novel technique to tackle the combinatorial LASSO problem with a hidden attribute and provide theoretical guarantees about the quality of the solution given a sufficient number of samples. Our method provably detects unfairness in the system. It should be noted that observing unfairness does not always imply that the designer of the system intended for such inequalities to arise. In such cases, our method acts as a check to detect and remove such unintended discrimination. While the current belief is that there is a trade-off between fairness and performance (Corbett-Davies et al., 2017; Kleinberg et al., 2017; Pleiss et al., 2017; Zliobaite, 2015; Zhao & Gordon, 2019), our

theoretical and experimental results show evidence on the contrary. Our theoretical results allow for a new understanding of fairness, as an "enabler" instead of as a "constraint".

**Contribution.** Broadly, we can categorize our contribution in the following points:

- **Defining the problem**: We formulate a novel combinatorial version of sparse linear regression which takes fairness/bias into the consideration. The addition of clustering comes at no extra cost in terms of the performance.
- **Invex relaxation**: Most of the current methods solve convex optimization problems as it makes the solution tractable. We propose a novel relaxation of the combinatorial problem and formally show that it is invex. To the best of our knowledge, this is the first invex relaxation for a combinatorial problem.
- **Theoretical Guarantees**: Our method can detect bias in the system. In particular, our method recovers the exact hidden attributes for each sample and thus provides an exact measure of bias between two different groups. Our method solves linear regression and clustering simultaneously with theoretical guarantees. To that end, we recover the true clusters (hidden attributes) and a regression parameter vector which is correct up to the sign of entries with respect to the true parameter vector. On a more technical side, we provide a primal-dual witness construction for our invex problem and provide theoretical guarantees for recovery. The sample complexity of our method varies logarithmically with respect to dimension of the regression parameter vector.

## 2 Notation and Problem Definition

In this section, we collect all the notations used throughout the paper. We also formally introduce our novel problem. We consider a problem where we have a binary hidden attribute, and where fairness depends upon the hidden attribute. Let $y \in \mathbb{R}$ be the response variable and $X \in \mathbb{R}^d$ be the observed attributes. Let $z^* \in \{-1, 1\}$ be the *hidden* attribute and $\gamma \in \mathbb{R}_{>0}$ be the amount of bias due to the hidden attribute. The response $y$ is generated using the following mechanism:

$$y = X^\intercal w^* + \gamma z^* + e \tag{1}$$

where $e$ is an independent noise term. For example, $y$ could represent the market salary of a new candidate, $X$ could represent the candidate's skills and $z$ could represent the population group the candidate belongs to (e.g., majority or minority). While the group of the candidate is not public knowledge, a bias associated with the candidate's group may be present in the underlying data. For our problem, we will assume that an estimate of the bias $\gamma \in \mathbb{R}_{>0}$ is available. In practice, even a rough estimate ($\pm 25\%$) of $\gamma$ also works well (See Appendix J).

Let $[d]$ denote the set $\{1, 2, \cdots, d\}$. We assume $X \in \mathbb{R}^d$ to be a zero mean sub-Gaussian random vector (Hsu et al., 2012) with covariance $\Sigma \in \mathbb{S}^d_+$, i.e., there exists a $\rho > 0$, such that for all $\alpha \in \mathbb{R}^d$ the following holds: $\mathbb{E}(\exp(\alpha^\intercal X)) \leqslant \exp(\frac{\|\alpha\|_2^2 \rho^2}{2})$. By simply taking $\alpha_i = r$ and $\alpha_k = 0, \forall k \neq i$, it follows that each entry of $X$ is sub-Gaussian with parameter $\rho$. In particular, we will assume that $\forall i \in [d], \frac{X_i}{\sqrt{\Sigma_{ii}}}$ is a sub-Gaussian random variable with parameter $\sigma > 0$. It follows trivially that $\max_{i \in [d]} \sqrt{\Sigma_{ii}} \sigma \leqslant \rho$. We will further assume that $e$ is zero mean independent sub-Gaussian noise with variance $\sigma_e$. We assume that as the number of samples increases, the noise in the model gently decreases. We model this by taking $\sigma_e = \frac{k}{\sqrt{\log n}}$ for some $k > 0$. Our setting works with a variety of random variables as the class of sub-Gaussian random variable includes for instance Gaussian variables, any bounded random variable (e.g., Bernoulli, multinomial, uniform), any random variable with strictly log-concave density, and any finite mixture of sub-Gaussian variables. Notice that for the group with $z = +1$, $\mathbb{E}(y) = \gamma$ and for the group with $z = -1$, $\mathbb{E}(y) = -\gamma$. This means that after correctly identifying groups, one could perform debiasing by subtracting or adding $\gamma$ for $z = +1$ and $-1$ respectively. After debiasing, the expected value of both groups would match (and be equal to 0). This complies with the notion of equal mean fairness proposed by Calders (2013).

The parameter vector $w^* \in \mathbb{R}^d$ is $s$-sparse, i.e., at most $s$ entries of $w^*$ are non-zero. We receive $n$ i.i.d. samples of $X \in \mathbb{R}^d$ and $y \in \mathbb{R}$ and collect them in $\mathbf{X} \in \mathbb{R}^{n \times d}$ and $\mathbf{y} \in \mathbb{R}^n$ respectively. Thus, in the finite-sample setting,

$$\mathbf{y} = \mathbf{X} w^* + \gamma \mathbf{z}^* + \mathbf{e}, \tag{2}$$

where $\mathbf{z}^* \in \{-1, 1\}^n$ and $\mathbf{e} \in \mathbb{R}^n$ both collect $n$ independent realizations of $z^* \in \{-1, 1\}$ and $e \in \mathbb{R}$. Our goal is to recover $w^*$ and $\mathbf{z}^*$ using the samples $(\mathbf{X}, \mathbf{y})$.

We denote a matrix $A \in \mathbb{R}^{p \times q}$ restricted to the columns and rows in $P \subseteq [p]$ and $Q \subseteq [q]$ respectively as $A_{PQ}$. Similarly, a vector $v \in \mathbb{R}^p$ restricted to entries in $P$ is denoted as $v_P$. We use $\mathrm{eig}_i(A)$ to denote the $i$-th eigenvalue (1st being the smallest) of matrix $A$. Similarly, $\mathrm{eig}_{\max}(A)$ denotes the maximum eigenvalue of matrix $A$. We use $\mathrm{diag}(A)$ to denote a vector containing the diagonal element of matrix $A$. By overriding the same notation, we use $\mathrm{diag}(v)$ to denote a diagonal matrix with its diagonal being the entries in vector $v$. We denote the inner product between two matrices $A$ and $B$ by $\langle A, B \rangle$, i.e., $\langle A, B \rangle = \mathrm{trace}(A^\mathsf{T} B)$, where $\mathrm{trace}$ denotes the trace of a matrix. The notation $A \succeq B$ denotes that $A - B$ is a positive semidefinite matrix. Similarly, $A \succ B$ denotes that $A - B$ is a positive definite matrix. For vectors, $\|v\|_p$ denotes the $\ell_p$-vector norm of vector $v \in \mathbb{R}^d$, i.e., $\|v\|_p = (\sum_{i=1}^d |v_i|^p)^{\frac{1}{p}}$. If $p = \infty$, then we define $\|v\|_\infty = \max_{i=1}^d |v_i|$. For matrices, $\|A\|_p$ denotes the induced $\ell_p$-matrix norm for matrix $A \in \mathbb{R}^{p \times q}$. In particular, $\|A\|_2$ denotes the spectral norm of $A$ and $\|A\|_\infty \triangleq \max_{i \in [p]} \sum_{j=1}^q |A_{ij}|$. A function $f(x)$ is of order $\Omega(g(x))$ and denoted by $f(x) = \Omega(g(x))$, if there exists a constant $C > 0$ such that for big enough $x_0$, $f(x) \geqslant Cg(x), \forall x \geqslant x_0$. Similarly, a function $f(x)$ is of order $\mathcal{O}(g(x))$ and denoted by $f(x) = \mathcal{O}(g(x))$, if there exists a constant $C > 0$ such that for big enough $x_0$, $f(x) \leqslant Cg(x), \forall x \geqslant x_0$. For brevity in our notations, we treat any quantity independent of $d, s$ and $n$ as constant. Detailed proofs for lemmas and theorems are available in the supplementary material.

## 3 Our New Optimization Problem and Invexity

In this section, we introduce our novel combinatorial problem and propose an invex relaxation. To the best of our knowledge, this is the first invex relaxation for a combinatorial problem. Without any information about the hidden attribute $\mathbf{z}^*$ in Equation (2), the following LASSO formulation could be incorrectly and unsuccessfully used to estimate the parameter $w^*$.

**Definition 1** (Standard LASSO).
$$\min_w \quad \frac{1}{n}(\mathbf{X}w - \mathbf{y})^\mathsf{T}(\mathbf{X}w - \mathbf{y}) + \lambda_n \|w\|_1 \tag{3}$$

However, without including $\mathbf{z}^*$, standard LASSO does not provide accurate estimation of $w^*$ in Equation (2). We provide the following novel formulation of LASSO which fits our goals of estimating both $w^*$ and $z^*$:

**Definition 2** (Combinatorial Fair LASSO).
$$\min_{w,\mathbf{z}} \quad \frac{1}{n}(\mathbf{X}w + \gamma\mathbf{z} - \mathbf{y})^\mathsf{T}(\mathbf{X}w + \gamma\mathbf{z} - \mathbf{y}) + \lambda_n \|w\|_1, \quad \text{such that } \mathbf{z}_i \in \{-1, 1\}, \ \forall i \in [n], \tag{4}$$
*where $\lambda_n > 0$ is the regularization level which depends on $n$.*

In its current form, optimization problem (4) is a non-convex mixed integer quadratic program (MIQP). Solving MIQP is NP-hard (See Appendix B). Next, we will provide a continuous but still non-convex relaxation of (4). For ease of notation, we define the following quantities:

$$l(w) \triangleq \frac{1}{n}(\mathbf{X}w - \mathbf{y})^\mathsf{T}(\mathbf{X}w - \mathbf{y}), \ \mathbf{Z} \triangleq \begin{bmatrix} 1 & \mathbf{z}^\mathsf{T} \\ \mathbf{z} & \mathbf{z}\mathbf{z}^\mathsf{T} \end{bmatrix}, \ \mathbf{M}(w) \triangleq \begin{bmatrix} l(w) & \frac{\gamma}{n}(\mathbf{X}w - \mathbf{y})^\mathsf{T} \\ \frac{\gamma}{n}(\mathbf{X}w - \mathbf{y}) & \frac{\gamma^2}{n}\mathbf{I}_{n \times n} \end{bmatrix},$$
$$\tag{5}$$

where $\mathbf{I}$ is an $n \times n$ identity matrix. We provide the following invex relaxation to the optimization problem (4).

**Definition 3** (Invex Fair LASSO).
$$\min_{w,\mathbf{Z}} \quad \langle \mathbf{M}(w), \mathbf{Z} \rangle + \lambda_n \|w\|_1, \quad \text{such that} \quad \mathrm{diag}(\mathbf{Z}) = \mathbf{1}, \ \mathbf{Z} \succeq \mathbf{0}_{n+1 \times n+1}. \tag{6}$$

Note that optimization problem (6) is continuous and convex with respect to $w$ and $\mathbf{Z}$ separately but it is not jointly convex (See Appendix N for details). Specifically, for a fixed $w$, the matrix $\mathbf{M}(w)$ becomes a constant and problem (6) resembles a semidefinite program. For a fixed $\mathbf{Z}$, problem (6) resembles a standard LASSO. Unfortunately, problem (6) is not jointly convex on $w$ and $\mathbf{Z}$, and thus, it might still remain difficult to solve. Next, we will provide arguments that despite being non-convex, optimization problem (6) belongs to a particular class of non-convex functions namely "invex" functions. We define "invexity" of functions, as a generalization of convexity (Hanson, 1981).

**Definition 4** (Invex function). *Let $\phi(t)$ be a function defined on a set $C$. Let $\eta$ be a vector valued function defined in $C \times C$ such that $\eta(t_1, t_2)^\mathsf{T} \nabla \phi(t_2)$, is well defined $\forall t_1, t_2 \in C$. Then, $\phi(t)$ is a $\eta$-invex function if $\phi(t_1) - \phi(t_2) \geqslant \eta(t_1, t_2)^\mathsf{T} \nabla \phi(t_2)$, $\forall t_1, t_2 \in C$.*

Note that convex functions are $\eta$-invex for $\eta(t_1, t_2) = t_1 - t_2$. Hanson (1981) showed that if the objective function and constraints are both $\eta$-invex with respect to same $\eta$ defined in $C \times C$, then Karush-Kuhn-Tucker (KKT) conditions are sufficient for optimality, while it is well-known that KKT conditions are necessary. Ben-Israel & Mond (1986) showed a function is invex if and only if each of its stationarity point is a global minimum.

In the next lemma, we show that the relaxed optimization problem (6) is indeed $\eta$-invex for a particular $\eta$ defined in $C \times C$ and a well defined set $C$. Before that, we will reformulate it into an equivalent optimization problem. Note that in the optimization problem (6), $\mathrm{diag}(\mathbf{Z}) = \mathbf{1}$. Thus, $\langle \mathbf{I}, \mathbf{Z} \rangle$ is a constant equal to $n + 1$. Using this, we can rewrite the optimization problem as:

$$\min_{w, \mathbf{Z}} \quad \langle \mathbf{M}(w), \mathbf{Z} \rangle + \lambda_n \|w\|_1 + \langle \mathbf{I}, \mathbf{Z} \rangle, \quad \text{such that} \quad \mathrm{diag}(\mathbf{Z}) = \mathbf{1}, \ \mathbf{Z} \succeq \mathbf{0}_{n+1 \times n+1}, \quad (7)$$

Let $C = \{(w, \mathbf{Z}) \mid w \in \mathbb{R}^d, \mathrm{diag}(\mathbf{Z}) = \mathbf{1}, \mathbf{Z} \succeq \mathbf{0}_{n+1 \times n+1}\}$. We take $\mathbf{M}'(w) = \mathbf{M}(w) + \mathbf{I}$ and the corresponding optimization problem becomes: $\min_{(w, \mathbf{Z}) \in C} \langle \mathbf{M}'(w), \mathbf{Z} \rangle + \lambda_n \|w\|_1$. We will show that $\forall (w, \mathbf{Z}) \in C, \langle \mathbf{M}'(w), \mathbf{Z} \rangle + \lambda_n \|w\|_1$ is an invex function. Note that by definition of the $\ell_1$-norm, $\|w\|_1 = \sup_{\|a\|_\infty = 1} \langle a, w \rangle$. Thus, it suffices to show that $\forall a \in \mathbb{R}^d, \langle \mathbf{M}'(w), \mathbf{Z} \rangle$ and $\langle a, w \rangle$ are invex for the same $\eta(w, \bar{w}, \mathbf{Z}, \bar{\mathbf{Z}})$.

**Lemma 1.** *For $(w, \mathbf{Z}) \in C$, the functions $f(w, \mathbf{Z}) = \langle \mathbf{M}'(w), \mathbf{Z} \rangle$ and $g(w, \mathbf{Z}) = \langle a, w \rangle$ are $\eta$-invex for $\eta(w, \bar{w}, \mathbf{Z}, \bar{\mathbf{Z}}) \triangleq \begin{bmatrix} w - \bar{w} \\ \mathbf{M}'(\bar{w})^{-1} \mathbf{M}'(w)(\mathbf{Z} - \bar{\mathbf{Z}}) \end{bmatrix}$, where we abuse the vector/matrix notation for clarity of presentation, and avoid the vectorization of matrices.*

Now that we have established that optimization problem (6) is invex, we are ready to discuss our main results in the next section.

# 4 Our Theoretical Analysis

In this section, we show that our Invex Fair Lasso formulation correctly recovers the hidden attributes and the regression parameter vector. More formally, we want to achieve the two goals by solving optimization problem (6) efficiently. First, we want to correctly and uniquely determine the hidden sensitive attribute for each data point, i.e., $\mathbf{z}^* \in \{-1, 1\}^n$. Second, we want to recover regression parameter vector which is close to the true parameter vector $w^* \in \mathbb{R}^d$ in $\ell_2$-norm. Let $\tilde{w}$ and $\tilde{\mathbf{Z}}$ be the solution to optimization problem (6). Then, we will prove that $\tilde{w}$ and $w^*$ have the same support and $\tilde{\mathbf{z}}$ constructed from $\tilde{\mathbf{Z}}$ is exactly equal to $\mathbf{z}^*$. We define $\Delta \triangleq (\tilde{w} - w^*)$.

## 4.1 KKT conditions

We start by writing the KKT conditions for optimization problem (6). Let $\mu \in \mathbb{R}^{n+1}$ and $\Lambda \succeq \mathbf{0}_{n+1 \times n+1}$ be the dual variables for optimization problem (6). For a fixed $\lambda_n$, the Lagrangian $L(w, \mathbf{Z}; \mu, \Lambda)$ can be written as $L(w, \mathbf{Z}; \mu, \Lambda) = \langle \mathbf{M}(w), \mathbf{Z} \rangle + \lambda_n \|w\|_1 + \langle \mathrm{diag}(\mu), \mathbf{Z} \rangle - \mathbf{1}^\mathsf{T} \mu - \langle \Lambda, \mathbf{Z} \rangle$. Using this Lagrangian, the KKT conditions at the optimum can be written as:

1. Stationarity conditions:

$$\frac{\partial \langle \mathbf{M}(w), \mathbf{Z} \rangle}{\partial w} + \lambda_n \mathbf{g} = \mathbf{0}_{d \times 1}, \quad (8)$$

where $\mathbf{g}$ is an element of the subgradient set of $\|w\|_1$, i.e., $\mathbf{g} \in \frac{\partial \|w\|_1}{\partial w}$ and $\|\mathbf{g}\|_\infty \leqslant 1$.

$$\mathbf{M}(w) + \mathrm{diag}(\mu) - \Lambda = \mathbf{0}_{n+1 \times n+1} \quad (9)$$

2. Complementary Slackness condition:

$$\langle \Lambda, \mathbf{Z} \rangle = 0 \quad (10)$$

3. Dual Feasibility condition:

$$\Lambda \succeq \mathbf{0}_{n+1 \times n+1} \qquad (11)$$

4. Primal Feasibility conditions:

$$w \in \mathbb{R}^d, \ \mathrm{diag}(\mathbf{Z}) = \mathbf{1}, \ \mathbf{Z} \succeq \mathbf{0}_{n+1 \times n+1} \qquad (12)$$

Next, we will provide a setting for primal and dual variables which satisfies all the KKT conditions. But before that, we will describe a set of technical assumptions which will help us in our analysis.

## 4.2 Assumptions

Let $S$ denote the support of $w^*$, i.e., $S = \{i \mid w_i^* \neq 0, \ i \in [d]\}$. Similarly, we define the complement of support $S$ as $S^c = \{i \mid w_i^* = 0, \ i \in [d]\}$. Let $|S| = s$ and $|S^c| = d - s$. For ease of notation, we define $\mathbf{H} \triangleq \mathbb{E}(XX^\intercal)$ and $\hat{\mathbf{H}} \triangleq \frac{1}{n}\mathbf{X}^\intercal\mathbf{X}$. As the first assumption, we need the minimum eigenvalue of the population covariance matrix of $X$ restricted to rows and columns in $S$ to be greater than zero. Later, we will show that this assumption is needed to uniquely recover $w$ in the optimization problem (6).

**Assumption 1** (Positive Definiteness of Hessian). $\mathbf{H}_{SS} > \mathbf{0}_{s \times s}$ *or equivalently* $\mathrm{eig}_{\min}(\mathbf{H}_{SS}) = C_{\min} > 0$.

In practice, we only deal with finite samples and not populations. In the next lemma, we will show that with a sufficient number of samples, a condition similar to Assumption 1 holds with high probability in the finite-sample setting.

**Lemma 2.** *If Assumption 1 holds and* $n = \Omega(\frac{s + \log d}{C_{\min}^2})$*, then* $\mathrm{eig}_{\min}(\hat{\mathbf{H}}_{SS}) \geqslant \frac{C_{\min}}{2}$ *with probability at least* $1 - \mathcal{O}(\frac{1}{d})$*.*

As the second assumption, we will need to ensure that the variates outside the support of $w^*$ do not exert lot of influence on the variates in the support of $w^*$. This sort of technical condition, known as the mutual incoherence condition, has been previously used in many problems related to regularized regression such as compressed sensing (Wainwright, 2009), Markov random fields (Ravikumar et al., 2010), non-parametric regression (Ravikumar et al., 2007), diffusion networks (Daneshmand et al., 2014), among others. We formally present this technical condition in what follows.

**Assumption 2** (Mutual Incoherence). $\|\mathbf{H}_{S^cS}\mathbf{H}_{SS}^{-1}\|_\infty \leqslant 1 - \alpha$ *for some* $\alpha \in (0, 1]$*.*

Again, we will show that with a sufficient number of samples, a condition similar to Assumption 2 holds in the finite-sample setting with high probability.

**Lemma 3.** *If Assumption 2 holds and* $n = \Omega(\frac{s^3(\log s + \log d)}{\tau(C_{\min}, \alpha, \sigma, \Sigma)})$*, then* $\|\hat{\mathbf{H}}_{S^cS}\hat{\mathbf{H}}_{SS}^{-1}\|_\infty \leqslant 1 - \frac{\alpha}{2}$ *with probability at least* $1 - \mathcal{O}(\frac{1}{d})$ *where* $\tau(C_{\min}, \alpha, \sigma, \Sigma)$ *is a constant independent of* $n, d$ *and* $s$*.*

In Appendix L, we experimentally show that Assumption 1 is easier to hold (i.e., $n \in \Omega(s + \log d)$) than Assumption 2 (i.e., $n \in \Omega(s^3 \log d)$). Eventually, both assumptions hold as the number of samples increases.

## 4.3 Construction of Primal and Dual Witnesses

In this subsection, we will provide a construction of primal and dual variables which satisfies the KKT conditions for optimization problem (6). To that end, we provide our first main result.

**Theorem 1** (Primal Dual Witness Construction). *If Assumptions 1 and 2 hold,* $\lambda_n \geqslant \frac{128\rho k}{\alpha}\frac{\sqrt{\log d}}{n}$ *and* $n = \Omega(\frac{s^3 \log d}{\tau_0(C_{\min}, \alpha, \sigma, \Sigma, \rho, k, \gamma)})$*, then the following setting of primal and dual variables*

*Primal Variables:* $\quad \tilde{w} = (\tilde{w}_S, \mathbf{0}_{d-s \times 1})$

*where,* $\tilde{w}_S = \arg\min_{w_S} \frac{1}{n}(\mathbf{X}_{.S}w_S + \gamma\mathbf{z}^* - \mathbf{y})^\intercal(\mathbf{X}_{.S}w_S + \gamma\mathbf{z}^* - \mathbf{y}) + \lambda_n\|w_S\|_1$

$$\mathbf{Z} = \mathbf{Z}^* \triangleq \begin{bmatrix} 1 & \mathbf{z}^{*\intercal} \\ \mathbf{z}^* & \mathbf{z}^*\mathbf{z}^{*\intercal} \end{bmatrix} \qquad (13)$$

*Dual Variables:* $\quad \mu = -\mathrm{diag}(M(\tilde{w})\mathbf{Z}^*), \quad \Lambda = M(\tilde{w}) - \mathrm{diag}(M(\tilde{w})\mathbf{Z}^*)$

*satisfies all the KKT conditions for optimization problem* (6) *with probability at least* $1 - \mathcal{O}(\frac{1}{n})$, *where* $\tau_0(C_{\min}, \alpha, \sigma, \Sigma, \rho, k, \gamma)$ *is a constant independent of* $s, d$ *and* $n$ *and thus, the primal variables are a globally optimal solution for* (6). *Furthermore, the above solution is also unique.*

**Proof Sketch.** The main idea behind our proofs is to verify that the setting of primal and dual variables in Theorem 1 satisfies all the KKT conditions described in subsection 4.1. We do this by proving multiple lemmas in subsequent subsections. The outline of the proof is as follows:

- It can be trivially verified that the primal feasibility condition (12) holds. Similarly, the second stationarity condition (9) holds by construction of $\Lambda$.
- In subsection 4.4, we use Lemmas 4 and 11 to verify that the stationarity condition (8) holds.
- In subsection 4.5, we use Lemma 5 to verify the complementary slackness condition (10).
- In subsection 4.6, we show that the dual feasibility condition (11) is satisfied using results from Lemmas 6, 7, 8 and 12.
- Finally, in subsection 4.7, we show that our proposed solution is also unique.

### 4.4 Verifying the Stationarity Condition (8)

In this subsection, we will show that the setting of $\tilde{w}$ and $\mathbf{Z}^*$ satisfies the first stationarity condition (8) by proving the following lemma.

**Lemma 4.** *If Assumptions 1 and 2 hold,* $\lambda_n \geqslant \frac{128\rho k}{\alpha}\frac{\sqrt{\log d}}{n}$ *and* $n = \Omega(\frac{s^3 \log d}{\tau_1(C_{\min}, \alpha, \sigma, \Sigma, \rho)})$, *then the setting of* $w$ *and* $\mathbf{Z}$ *from equation* (13) *satisfies the stationarity condition* (8) *with probability at least* $1 - \mathcal{O}(\frac{1}{d})$, *where* $\tau_1(C_{\min}, \alpha, \sigma, \Sigma, \rho)$ *is a constant independent of* $d, s$ *or* $n$.

### 4.5 Verifying the Complementary Slackness (10)

Next, we will show that the setting of $\Lambda$ and $\mathbf{Z}$ in (13) satisfies the complementary slackness condition (10). To this end, we will show the following:

**Lemma 5.** *Let* $\Lambda$ *be defined as in equation* (13), *then* $\zeta^* \triangleq \begin{bmatrix} 1 \\ \mathbf{z}^* \end{bmatrix}$ *is an eigenvector of* $\Lambda$ *corresponding to the eigenvalue* $0$. *Furthermore,* $\langle \Lambda, \mathbf{Z}^* \rangle = 0$.

*Proof.* We will show that $\Lambda \zeta^* = \mathbf{0}_{n+1 \times 1}$. Note that,

$$\Lambda = M(w) - \text{diag}(M(w)\mathbf{Z}^*) = \begin{bmatrix} -\frac{\gamma}{n}(Xw - y)^T\mathbf{z}^* & \frac{\gamma}{n}(Xw - y)^T \\ \frac{\gamma}{n}(Xw - y) & -\text{diag}(\frac{\gamma}{n}(Xw - y)\mathbf{z}^{*T}) \end{bmatrix}$$

Multiplying the above matrix with $\zeta^*$ gives us $\mathbf{0}_{n+1 \times 1}$. Now $\langle \Lambda, \mathbf{Z}^* \rangle = \text{trace}(\Lambda^{\intercal}\mathbf{Z}^*) = \text{trace}(\Lambda \zeta^* \zeta^{*\intercal}) = 0$ as $\Lambda \zeta^* = \mathbf{0}_{n+1 \times 1}$. $\qquad\square$

### 4.6 Verifying the Dual Feasibility (11)

We have already shown that $\Lambda$ has $0$ as one of its eigenvalues. To verify that it satisfies the dual feasibility condition (11), we show that second minimum eigenvalue of $\Lambda$ is greater than zero with high probability. At this point, it might not be clear why strict positivity is necessary, but this will be argued later in subsection 4.7. Now, note that:

$$\mathbb{P}(\text{eig}_2(\Lambda) > 0) \geqslant \mathbb{P}(\text{eig}_2(\Lambda) > 0, \|\Delta\|_2 \leqslant h(n)) \geqslant \mathbb{P}(\text{eig}_2(\Lambda) > 0 | \|\Delta\|_2 \leqslant h(n))\mathbb{P}(\|\Delta\|_2 \leqslant h(n)) \tag{14}$$

where $h(n)$ is a function of $n$. We bound $\mathbb{P}(\text{eig}_2(\Lambda) > 0)$ in two parts. First, we bound $\mathbb{P}(\text{eig}_2(\Lambda) > 0)$ given that $\|\Delta\|_2 \leqslant h(n)$ and then we bound the probability of $\|\Delta\|_2 \leqslant h(n)$.

**Lemma 6.** *Given that* $\|\Delta\|_2 \leqslant h(n)$, *the second minimum eigenvalue of* $\Lambda$ *as defined in equation* (13) *is strictly greater than* $0$ *with probability at least* $1 - \exp(-\frac{\gamma^2}{8(\rho^2 h(n)^2 + \sigma_e^2)} + \log n)$.

*Proof.* As the first step, we invoke Haynesworth's inertia additivity formula (Haynsworth, 1968) to prove our claim. Let $R$ be a block matrix of the form $R = \begin{bmatrix} A & B \\ B^{\intercal} & C \end{bmatrix}$, then inertia of matrix $R$,

denoted by $\text{In}(R)$, is defined as the tuple $(\pi(R), \nu(R), \delta(R))$ where $\pi(R)$ is the number of positive eigenvalues, $\nu(R)$ is the number of negative eigenvalues and $\delta(R)$ is the number of zero eigenvalues of matrix $R$. Haynesworth's inertia additivity formula is given as:

$$\text{In}(R) = \text{In}(C) + \text{In}(A - B^\mathsf{T} C^{-1} B) \tag{15}$$

Note that,

$$\Lambda = \mathbf{M}(w) - \text{diag}(\mathbf{M}(w)\mathbf{Z}^*) = \begin{bmatrix} -\frac{\gamma}{n}(\mathbf{X}w - \mathbf{y})^\mathsf{T}\mathbf{z}^* & \frac{\gamma}{n}(\mathbf{X}w - \mathbf{y})^\mathsf{T} \\ \frac{\gamma}{n}(\mathbf{X}w - \mathbf{y}) & -\text{diag}(\frac{\gamma}{n}(\mathbf{X}w - \mathbf{y})\mathbf{z}^{*\mathsf{T}}) \end{bmatrix}$$

Then the following holds true by applying equation (15):

$$\text{In}(\Lambda) = \text{In}(-\text{diag}(\frac{\gamma}{n}(\mathbf{X}w - \mathbf{y})\mathbf{z}^{*\mathsf{T}})) + \text{In}(-\frac{\gamma}{n}(\mathbf{X}w - \mathbf{y})^\mathsf{T}\mathbf{z}^* - \frac{\gamma}{n}(\mathbf{X}w - \mathbf{y})^\mathsf{T}$$
$$(-\text{diag}(\frac{\gamma}{n}(\mathbf{X}w - \mathbf{y})\mathbf{z}^{*\mathsf{T}})^{-1}\frac{\gamma}{n}(\mathbf{X}w - \mathbf{y}))$$

Notice that the term $-\frac{\gamma}{n}(\mathbf{X}w - \mathbf{y})^\mathsf{T}\mathbf{z}^* - \frac{\gamma}{n}(\mathbf{X}w - \mathbf{y})^\mathsf{T}(-\text{diag}(\frac{\gamma}{n}(\mathbf{X}w - \mathbf{y})\mathbf{z}^{*\mathsf{T}})^{-1}\frac{\gamma}{n}(\mathbf{X}w - \mathbf{y}))$ evaluates to 0. Thus, it has 0 positive eigenvalue, 0 negative eigenvalue and 1 zero eigenvalue. We have also shown in Lemma 5 that $\Lambda$ has at least 1 zero eigenvalue. It follows that

$$\pi(\Lambda) = \pi(-\text{diag}(\frac{\gamma}{n}(\mathbf{X}w - \mathbf{y})\mathbf{z}^{*\mathsf{T}})), \quad \nu(\Lambda) = \nu(-\text{diag}(\frac{\gamma}{n}(\mathbf{X}w - \mathbf{y})\mathbf{z}^{*\mathsf{T}}))$$
$$\delta(\Lambda) = \delta(-\text{diag}(\frac{\gamma}{n}(\mathbf{X}w - \mathbf{y})\mathbf{z}^{*\mathsf{T}})) + 1 \tag{16}$$

Next, we will show that $-\text{diag}(\frac{\gamma}{n}(\mathbf{X}w - \mathbf{y})\mathbf{z}^{*\mathsf{T}})$ has all of its eigenvalues being positive.

**Lemma 7.** *For a given $\|\Delta\|_2 \leqslant h(n)$, all eigenvalues of $-\text{diag}(\frac{\gamma}{n}(\mathbf{X}w - \mathbf{y})\mathbf{z}^{*\mathsf{T}})$ are strictly greater than 0 with probability at least $1 - \exp(-\frac{\gamma^2}{8(\rho^2 h(n)^2 + \sigma_e^2)} + \log n)$.*

*Proof.* Using equation (1), we can expand the term $-\text{diag}(\frac{\gamma}{n}(\mathbf{X}w - \mathbf{y})\mathbf{z}^{*\mathsf{T}})$ as $-\text{diag}(\frac{\gamma}{n}(\mathbf{X}(w - w^*) - \gamma\mathbf{z}^* - \mathbf{e})\mathbf{z}^{*\mathsf{T}})$. Since eigenvalues of a diagonal matrix are its diagonal elements, we focus on the $i$-th diagonal element of $-\text{diag}(\frac{\gamma}{n}(\mathbf{X}\Delta - \gamma\mathbf{z}^* - \mathbf{e})\mathbf{z}^{*\mathsf{T}})$ which is $\frac{\gamma^2}{n} - \frac{\gamma}{n}z_i^*(\mathbf{X}_{i\cdot}^\mathsf{T}\Delta + \mathbf{e}_i)$. Note that $(\mathbf{X}_{i\cdot}^\mathsf{T}\Delta + \mathbf{e}_i)$ is a sub-Gaussian random variable with parameter $\rho^2\|\Delta\|_2^2 + \sigma_e^2$. Using the tail inequality for sub-Gaussian random variables, for some $t > 0$, we can write:

$$\mathbb{P}((\mathbf{X}_{i\cdot}\Delta + \mathbf{e}_i) \geqslant t) \leqslant \exp(-\frac{t^2}{2(\rho^2\|\Delta\|_2^2 + \sigma_e^2)})$$

We take union bound across all the diagonal elements and replace $t = \frac{\gamma}{2}$ and $\|\Delta\|_2 \leqslant h(n)$ to complete the proof, i.e.,

$$\mathbb{P}(\exists i \in [n], (\mathbf{X}_{i\cdot}\Delta + \mathbf{e}_i) \geqslant t) \leqslant n\exp(-\frac{\gamma^2}{8(\rho^2 h(n)^2 + \sigma_e^2)}). \tag{17}$$

$\square$

The result of Lemma 6 follows directly from Lemma 7 and equation (16). $\square$

Now, we are ready to bound $\|\Delta\|_2$. Due to our primal dual construction, $\|\Delta\|_2$ is simply equal to $\|\Delta_S\|_2$. We provide a bound on $\Delta_S$ in the following lemma:

**Lemma 8.** *If Assumptions 1 and 2 hold, $\lambda_n \geqslant \frac{128\rho k\sqrt{\log d}}{\alpha n}$ and $n = \Omega(\frac{s^3 \log d}{\tau_2(C_{\min}, \rho, k)})$, then $\|\Delta_S\|_2 \leqslant \frac{2\lambda_n\sqrt{s}}{C_{\min}}$ with probability at least $1 - \mathcal{O}(\frac{1}{d})$ where $\tau_2(C_{\min}, \rho, k)$ is a constant independent of $s, d$ or $n$.*

By taking $h(n) = \frac{2\lambda_n\sqrt{s}}{C_{\min}}$ in (14), we get the following: $\mathbb{P}(\text{eig}_2(\Lambda) > 0) \geqslant 1 - \mathcal{O}(\frac{1}{n})$, as long as $n = \Omega(\frac{s^3 \log d}{\tau_3(C_{\min}, \rho, k, \alpha, \gamma)})$, where $\tau_3(C_{\min}, \rho, k, \alpha, \gamma)$ is a constant independent of $s, d$ and $n$. The above results combined with the property that optimization problem (6) is invex ensure that the setting of primal and dual variables in Theorem 1 is indeed the globally optimal solution to the problem (6). It remains to show that this solution is also unique.

## 4.7 Uniqueness of the Solution

First, we prove that $\mathbf{Z}^*$ is a unique solution. Suppose there is another solution $\bar{\mathbf{Z}}$ which satisfies all KKT conditions and is optimal. Then, $\bar{\mathbf{Z}} \succeq \mathbf{0}_{n+1 \times n+1}$ and $\langle \Lambda, \bar{\mathbf{Z}} \rangle = 0$. Since, $\Lambda \succeq \mathbf{0}_{n+1 \times n+1}$ and $\mathrm{eig}_2(\Lambda) > 0$, $\zeta^*$ spans all of its null space. This enforces that $\bar{\mathbf{Z}}$ is a multiple of $\mathbf{Z}^*$. But primal feasibility dictates that $\mathrm{diag}(\bar{\mathbf{Z}}) = \mathbf{1}$. It follows that $\bar{\mathbf{Z}} = \mathbf{Z}^*$. To show that $\tilde{w}$ is unique, it suffices to show that $\tilde{w}_S$ is unique. After substituting $\mathbf{Z} = \mathbf{Z}^*$, we observe that the Hessian of optimization problem (6) with respect to $w$ and restricted to rows and columns in $S$, i.e., $\hat{\mathbf{H}}_{SS}$ is positive definite. This ensures that $\tilde{w}$ is a unique solution.

The setting of primal and dual variables in Theorem 1 not only solves the optimization problem (6) but also gives rise to the following results:

**Corollary 1.** *If Assumptions 1 and 2 hold, $\lambda_n \geq \frac{128\rho k}{\alpha} \frac{\sqrt{\log d}}{n}$ and $n = \Omega(\frac{s^3 \log d}{\tau_1(C_{\min}, \alpha, \sigma, \Sigma, \rho)})$, then the following statements are true with probability at least $1 - \mathcal{O}(\frac{1}{n})$:*

1. *The solution $\mathbf{Z}$ correctly recovers hidden attribute for each sample, i.e., $\mathbf{Z} = \mathbf{Z}^* = \zeta^* \zeta^{*\mathsf{T}}$.*
2. *The support of recovered regression parameter $\tilde{w}$ matches exactly with the support of $w^*$.*
3. *If $\min_{i \in S} |w_i^*| \geq \frac{4\lambda_n \sqrt{s}}{C_{\min}}$ then for all $i \in [d]$, $\tilde{w}_i$ and $w_i^*$ match up to their sign.*

# 5 Experimental Validation

**Synthetic Experiments.** We validate our theoretical result in Theorem 1 and Corollary 1 by conducting experiments on synthetic data. We show that for a fixed $s$, we need $n = 10^\beta \log d$ samples for recovering the exact support of $w^*$ and exact hidden attributes $\mathbf{Z}^*$, where $\beta \equiv \beta(s, C_{\min}, \alpha, \sigma, \Sigma, \rho, \gamma, k)$ is a control parameter which is independent of $d$. We draw $\mathbf{X} \in \mathbb{R}^{n \times d}$ and $\mathbf{e} \in \mathbb{R}^n$ from Gaussian distributions. We randomly generate $w^* \in \mathbb{R}^d$ with $s = 10$ non-zero entries. Regarding the hidden attribute $\mathbf{z}^* \in \{-1, 1\}^n$, we set $\frac{n}{2}$ entries as $+1$ and the rest as $-1$. The response $\mathbf{y} \in \mathbb{R}^n$ is generated according to (1). According to Theorem 1, the regularizer $\lambda_n$ is chosen to be equal to $\frac{128\rho k}{\alpha} \frac{\sqrt{\log d}}{n}$. We solve optimization problem (6) by using an alternate optimization algorithm that converges to the optimal solution (See Appendix K for details). Figure 2a shows that our method recovers the true support as we increase the number of samples. Similarly, Figure 2c shows that as the number of samples increase, our recovered hidden attributes are 100% correct. Curves line up perfectly in Figure 2b and 2d when plotting with respect to the control parameter $\beta = \log \frac{n}{\log d}$. This validates our theoretical results (Details in Appendix M).

**Real World Experiments.** We show applicability of our method by identify groups with bias in the Communities and Crime data set (Redmond, 2002) and the Student Performance data set (Cortez, 2008). In both cases, our method is able to recover groups with bias (Details in Appendix O).

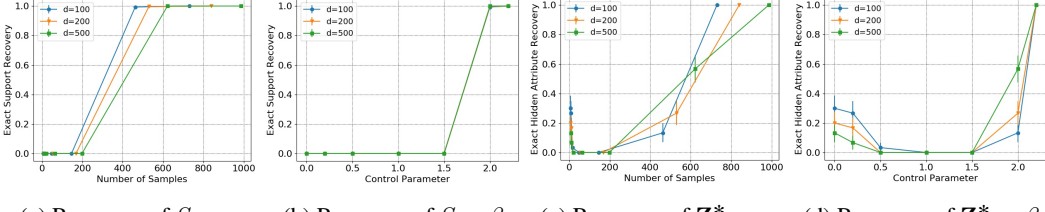

| (a) Recovery of $S$ vs $n$ | (b) Recovery of $S$ vs $\beta$ | (c) Recovery of $\mathbf{Z}^*$ vs $n$ | (d) Recovery of $\mathbf{Z}^*$ vs $\beta$ |

Figure 2: Left two: Exact support recovery of $w^*$ across 30 runs. Right two: Exact hidden attribute recovery of $\mathbf{Z}^*$ across 30 runs.

# 6 Concluding Remarks

In this paper, we provide a novel formulation of invex fair LASSO which incorporates fairness constraints into the standard LASSO problem without compromising on the performance. We show that invexity of our optimization problem allows for a tractable solution. We provide provable

theoretical guarantees for our solution and further validate them by computational experiments. The sample complexity of our method is polynomial in terms of sparsity and logarithmic in terms of the dimension of the true parameter. Our method helps to identify and subsequently remove bias in the sparse regression model. In the future, it will be interesting to study invex relaxations of other models of fairness. Since set of invex functions subsumes convex functions, invexity will enable us to tackle a larger set of problems.

## Societal Impact And Limitations

Fairness in machine learning is an active field of research. As it has the potential to affect the basic well being of our society, it should always be used with caution. This is especially important when one moves away from the theoretical setting and tries to apply fairness algorithms to real world data where the validity of technical assumptions cannot be easily verified. As with any algorithm in prior literature (Calders, 2013; Agarwal, 2019; Fitzsimons, 2019; Berk, 2017; Chzhen et al., 2020), one has to be extra cautious when interpreting the results of our method as well. In particular, we emphasize that our method does not characterize the nature of the recovered bias – good or bad. Our proposed hidden attribute could be seen as a proxy for algorithmic bias, but one that exists outside the features of the data itself. To interpret the meaning of such an attribute, we advise consulting with a domain expert, and to regress the discovered hidden attribute against some existing predictors provided by the expert. We also caution practitioners that such a notion of hidden bias would naturally not extend to every task, and should not be used as a silver bullet to justify fairness in socially significant contexts.

## Acknowledgments and Disclosure of Funding

This material is based upon work supported by the National Science Foundation under Grant No. 2134209-DMS.

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
