# Supplementary Material: Fair Sparse Regression with Clustering: An Invex Relaxation for a Combinatorial Problem

## A   Proof of Lemma 1

**Lemma 1**   *For $(w, \mathbf{Z}) \in C$, the functions $f(w, \mathbf{Z}) = \langle \mathbf{M}'(w), \mathbf{Z} \rangle$ and $g(w, \mathbf{Z}) = \langle a, w \rangle$ are $\eta$-invex for $\eta(w, \bar{w}, \mathbf{Z}, \bar{\mathbf{Z}}) \triangleq \begin{bmatrix} w - \bar{w} \\ \mathbf{M}'(\bar{w})^{-1}\mathbf{M}'(w)(\mathbf{Z} - \bar{\mathbf{Z}}) \end{bmatrix}$, where we abuse the vector/matrix notation for clarity of presentation, and avoid the vectorization of matrices.*

*Proof.*  We need to prove the following two inequalities.

$$f(w, \mathbf{Z}) - f(\bar{w}, \bar{\mathbf{Z}}) - \langle \nabla_{\bar{w}, \bar{\mathbf{Z}}} f(w, \mathbf{Z}), \eta(w, \bar{w}, \mathbf{Z}, \bar{\mathbf{Z}}) \rangle \geqslant 0 \,, \tag{18}$$

$$g(w, \mathbf{Z}) - g(\bar{w}, \bar{\mathbf{Z}}) - \langle \nabla_{\bar{w}, \bar{\mathbf{Z}}} g(w, \mathbf{Z}), \eta(w, \bar{w}, \mathbf{Z}, \bar{\mathbf{Z}}) \rangle \geqslant 0 \,. \tag{19}$$

First, we observe that function $g(w, \mathbf{Z})$ only depends on $w$ and moreover, $\forall a \in \mathbb{R}^d$, $g(w, \mathbf{Z})$ is convex in $w$. Thus, the inequality (19) holds trivially. Note that $f(w, \mathbf{Z}) = \langle \mathbf{M}'(w), \mathbf{Z} \rangle = \sum_{ij} \mathbf{M}'_{ij}(w) \mathbf{Z}_{ij}$. Then,

$$\frac{\partial f(\bar{w}, \bar{\mathbf{Z}})}{\partial w} = \sum_{ij} \bar{\mathbf{Z}}_{ij} \frac{\partial \mathbf{M}'_{ij}(\bar{w})}{\partial w}, \quad \frac{\partial f(\bar{w}, \bar{\mathbf{Z}})}{\partial \mathbf{Z}} = \mathbf{M}'(\bar{w})$$

We further note that the diagonal elements of $\mathbf{M}'(w)$ are convex with respect to $w$ and the off diagonal elements are linear. Therefore, we can write the following:

$$\mathbf{M}'_{ii}(w) - \mathbf{M}'_{ii}(\bar{w}) \geqslant \langle \frac{\partial \mathbf{M}'_{ii}(\bar{w})}{\partial w}, w - \bar{w} \rangle, \forall i \in [n + 1]$$

$$\mathbf{M}'_{ij}(w) - \mathbf{M}'_{ij}(\bar{w}) = \langle \frac{\partial \mathbf{M}'_{ij}(\bar{w})}{\partial w}, w - \bar{w} \rangle, \forall i, j \in [n + 1], i \neq j$$

Since $\bar{\mathbf{Z}}_{ii} \geqslant 0$, it follows that

$$\bar{\mathbf{Z}}_{ij} \langle \frac{\partial \mathbf{M}'_{ij}(\bar{w})}{\partial w}, w - \bar{w} \rangle \leqslant \bar{\mathbf{Z}}_{ij} (\mathbf{M}'_{ij}(w) - \mathbf{M}'_{ij}(\bar{w})) \,.$$

Now, we prove that $f(w, \mathbf{Z})$ is indeed $\eta$-invex, that is

$$f(w, \mathbf{Z}) - f(\bar{w}, \bar{\mathbf{Z}}) - \langle \nabla_{\bar{w}, \bar{\mathbf{Z}}} f(w, \mathbf{Z}), \eta(w, \bar{w}, \mathbf{Z}, \bar{\mathbf{Z}}) \rangle$$

$$= \langle \mathbf{M}'(w), \mathbf{Z} \rangle - \langle \mathbf{M}'(\bar{w}), \bar{\mathbf{Z}} \rangle - \langle \sum_{ij} \bar{\mathbf{Z}}_{ij} \frac{\partial \mathbf{M}'_{ij}(\bar{w})}{\partial w}, w - \bar{w} \rangle - \langle \mathbf{M}'(\bar{w}), \mathbf{M}'(\bar{w})^{-1} \mathbf{M}'(w)(\mathbf{Z} - \bar{\mathbf{Z}}) \rangle$$

$$\geqslant \langle \mathbf{M}'(w), \mathbf{Z} \rangle - \langle \mathbf{M}'(\bar{w}), \bar{\mathbf{Z}} \rangle - \sum_{ij} \bar{\mathbf{Z}}_{ij} (\mathbf{M}'_{ij}(w) - \mathbf{M}'_{ij}(\bar{w})) - \langle \mathbf{M}'(w), \mathbf{Z} \rangle + \langle \mathbf{M}'(w), \bar{\mathbf{Z}} \rangle$$

$$= 0$$

This proves that $f(w, \mathbf{Z})$ is $\eta$-invex in $(w, \mathbf{Z}) \in C$. $\qquad \square$

## B   Mixed Integer Quadratic Program (MIQP) (4) is NP-Hard

In this section, we will show that the MIQP presented in (4) is at least as hard to solve as a $0 - 1$ Quadratic Program. It should be noted that MIQP (4) is stated for a fixed $\mathbf{X}$. However, since the entries in $\mathbf{X}$ are drawn from a sub-Gaussian distribution, matrix $\mathbf{X}$ can potentially realize any real matrix in $\mathbb{R}^{n \times d}$.

**Lemma 9.** *The Mixed Integer Quadratic Program (MIQP)* (4) *is NP-hard.*

*Proof.* We will consider the case when $\lambda_n = 0$. Other cases will be at least as difficult as this case. First, we write optimization problem (4) in the following form:

$$
\min_{w \in \mathbb{R}^d, \mathbf{z} \in \{-1,1\}^n} \frac{1}{2} w^\mathsf{T} (\frac{2}{n} \mathbf{X}^\mathsf{T} \mathbf{X}) w + \mathbf{z}^\mathsf{T} \frac{2}{n} \gamma \mathbf{X} w + \frac{1}{2} \mathbf{z}^\mathsf{T} \frac{2}{n} \gamma^2 \mathbf{I} \mathbf{z} - \frac{2}{n} \mathbf{y}^\mathsf{T} \mathbf{X} w - \frac{2}{n} \gamma \mathbf{y}^\mathsf{T} \mathbf{z}
$$
$$
= \min_{\mathbf{z} \in \{-1,1\}^n} \left( \frac{1}{2} \mathbf{z}^\mathsf{T} \frac{2}{n} \gamma^2 \mathbf{I} \mathbf{z} - \frac{2}{n} \gamma \mathbf{y}^\mathsf{T} \mathbf{z} + \left( \min_{w \in \mathbb{R}^d} \frac{1}{2} w^\mathsf{T} (\frac{2}{n} \mathbf{X}^\mathsf{T} \mathbf{X}) w + (\mathbf{z}^\mathsf{T} \frac{2}{n} \gamma \mathbf{X} - \frac{2}{n} \mathbf{y}^\mathsf{T} \mathbf{X}) w \right) \right) \tag{20}
$$

We observe that $w = (\mathbf{X}^\mathsf{T} \mathbf{X})^\dagger \mathbf{X}^\mathsf{T} (-\gamma \mathbf{z} + \mathbf{y})$ solves the nested optimization problem, where $(\cdot)^\dagger$ denotes the pseudo-inverse. Thus, substituting the optimal value of $w$, we get the following optimization problem:

$$
\min_{\mathbf{z} \in \{-1,1\}^n} \frac{\gamma^2}{n} \mathbf{z}^\mathsf{T} (\mathbf{I} - \mathbf{X}(\mathbf{X}^\mathsf{T} \mathbf{X})^\dagger \mathbf{X}^\mathsf{T}) \mathbf{z} - \frac{2\gamma}{n} \mathbf{y}^\mathsf{T} (\mathbf{I} - \mathbf{X}(\mathbf{X}^\mathsf{T} \mathbf{X})^\dagger \mathbf{X}^\mathsf{T}) \mathbf{z} \tag{21}
$$

Observe that $I - \mathbf{X}(\mathbf{X}^\mathsf{T} \mathbf{X})^\dagger \mathbf{X}^\mathsf{T}$ can potentially be any fixed real matrix in $\mathbb{R}^{n \times n}$. By simply substituting $\mathbf{z}' = \frac{\mathbf{z}+1}{2}$, we get a $0 - 1$ Quadratic Program which is known to be NP-Hard (Billionnet, 2010). $\qquad \square$

## C   Proof of Lemma 2

**Lemma 2**  *If Assumption 1 holds and $n = \Omega(\frac{s + \log d}{C_{\min}^2})$, then $\mathrm{eig}_{\min}(\hat{\mathbf{H}}_{SS}) \geqslant \frac{C_{\min}}{2}$ with probability at least $1 - \mathcal{O}(\frac{1}{d})$.*

*Proof.* By the Courant-Fischer variational representation (Horn & Johnson, 2012):

$$
\mathrm{eig}_{\min}(\mathbb{E}(XX^\mathsf{T})_{SS}) = \min_{\|y\|_2=1} y^\mathsf{T} \mathbb{E}(XX^\mathsf{T})_{SS} y = \min_{\|y\|_2=1} y^\mathsf{T} (\mathbb{E}(XX^\mathsf{T})_{SS} - \frac{1}{n} \mathbf{X}_S^\mathsf{T} \mathbf{X}_S + \frac{1}{n} \mathbf{X}_S^\mathsf{T} \mathbf{X}_S) y
$$
$$
\leqslant y^\mathsf{T} (\mathbb{E}(XX^\mathsf{T})_{SS} - \frac{1}{n} \mathbf{X}_S^\mathsf{T} \mathbf{X}_S + \frac{1}{n} \mathbf{X}_S^\mathsf{T} \mathbf{X}_S) y
$$
$$
= y^\mathsf{T} (\mathbb{E}(XX^\mathsf{T})_{SS} - \frac{1}{n} \mathbf{X}_S^\mathsf{T} \mathbf{X}_S) y + y^\mathsf{T} \frac{1}{n} \mathbf{X}_S^\mathsf{T} \mathbf{X}_S y \tag{22}
$$

It follows that

$$
\mathrm{eig}_{\min}(\frac{1}{n} \mathbf{X}_S^\mathsf{T} \mathbf{X}_S) \geqslant C_{\min} - \|\mathbb{E}(XX^\mathsf{T})_{SS} - \frac{1}{n} \mathbf{X}_S^\mathsf{T} \mathbf{X}_S\|_2 \tag{23}
$$

The term $\|\mathbb{E}(XX^\mathsf{T})_{SS} - \frac{1}{n} \mathbf{X}_S^\mathsf{T} \mathbf{X}_S\|_2$ can be bounded using Proposition 2.1 in Vershynin (2012) for sub-Gaussian random variables. In particular,

$$
\mathbb{P}(\|\mathbb{E}(XX^\mathsf{T})_{SS} - \frac{1}{n} \mathbf{X}_S^\mathsf{T} \mathbf{X}_S\|_2 \geqslant \epsilon) \leqslant 2 \exp(-c\epsilon^2 n + s) \tag{24}
$$

for some constant $c > 0$. Taking $\epsilon = \frac{C_{\min}}{2}$, we show that $\mathrm{eig}_{\min}(\frac{1}{n} \mathbf{X}_S^\mathsf{T} \mathbf{X}_S) \geqslant \frac{C_{\min}}{2}$ with probability at least $1 - 2 \exp(-\frac{cC_{\min}^2 n}{4} + s)$. $\qquad \square$

## D   Proof of Lemma 3

**Lemma 3**  *If Assumption 2 holds and $n = \Omega(\frac{s^3(\log s + \log d)}{\tau(C_{\min}, \alpha, \sigma, \Sigma)})$, then $\|\hat{\mathbf{H}}_{S^c S} \hat{\mathbf{H}}_{SS}^{-1}\|_\infty \leqslant 1 - \frac{\alpha}{2}$ with probability at least $1 - \mathcal{O}(\frac{1}{d})$ where $\tau(C_{\min}, \alpha, \sigma, \Sigma)$ is a constant independent of $n, d$ and $s$.*

*Proof.* Before we prove the result of Lemma 3, we will prove a helper lemma.

**Lemma 10.** *If Assumption 2 holds then for some $\delta > 0$, the following inequalities hold:*

$$\mathbb{P}(\|\hat{\mathbf{H}}_{S^c S} - \mathbf{H}_{S^c S}\|_\infty \geqslant \delta) \leqslant 4(d-s)s \exp(-\frac{n\delta^2}{128s^2(1+4\sigma^2)\max_l \Sigma_{ll}^2})$$

$$\mathbb{P}(\|\hat{\mathbf{H}}_{SS} - \mathbf{H}_{SS}\|_\infty \geqslant \delta) \leqslant 4s^2 \exp(-\frac{n\delta^2}{128s^2(1+4\sigma^2)\max_l \Sigma_{ll}^2}) \tag{25}$$

$$\mathbb{P}(\|(\hat{\mathbf{H}}_{SS})^{-1} - (\mathbf{H}_{SS})^{-1}\|_\infty \geqslant \delta) \leqslant 2\exp(-\frac{c\delta^2 C_{\min}^4 n}{4s} + s) + 2\exp(-\frac{cC_{\min}^2 n}{4} + s)$$

*Proof.* Let $A_{ij}$ be $(i,j)$-th entry of $\hat{\mathbf{H}}_{S^c S} - \mathbf{H}_{S^c S}$. Clearly, $\mathbb{E}(A_{ij}) = 0$. By using the definition of the $\|\cdot\|_\infty$ norm, we can write:

$$\begin{aligned}
\mathbb{P}(\|\hat{\mathbf{H}}_{S^c S} - \mathbf{H}_{S^c S}\|_\infty \geqslant \delta) &= \mathbb{P}(\max_{i \in S^c} \sum_{j \in S} |A_{ij}| \geqslant \delta) \\
&\leqslant (d-s)\mathbb{P}(\sum_{j \in S} |A_{ij}| \geqslant \delta) \\
&\leqslant (d-s)s\mathbb{P}(|A_{ij}| \geqslant \frac{\delta}{s})
\end{aligned} \tag{26}$$

where the second last inequality comes as a result of the union bound across entries in $S^c$ and the last inequality is due to the union bound across entries in $S$. Recall that $X_i, i \in [d]$ are zero mean random variables with covariance $\Sigma$ and each $\frac{X_i}{\sqrt{\Sigma_{ii}}}$ is a sub-Gaussian random variable with parameter $\sigma$. Using the results from Lemma 1 of Ravikumar et al. (2011), for some $\delta \in (0, s\max_l \Sigma_{ll}8(1+4\sigma^2))$, we can write:

$$\mathbb{P}(|A_{ij}| \geqslant \frac{\delta}{s}) \leqslant 4\exp(-\frac{n\delta^2}{128s^2(1+4\sigma^2)\max_l \Sigma_{ll}^2}) \tag{27}$$

Therefore,

$$\mathbb{P}(\|\hat{\mathbf{H}}_{S^c S} - \mathbf{H}_{S^c S}\|_\infty \geqslant \delta) \leqslant 4(d-s)s \exp(-\frac{n\delta^2}{128s^2(1+4\sigma^2)\max_l \Sigma_{ll}^2}) \tag{28}$$

Similarly, we can show that

$$\mathbb{P}(\|\hat{\mathbf{H}}_{SS} - \mathbf{H}_{SS}\|_\infty \geqslant \delta) \leqslant 4s^2 \exp(-\frac{n\delta^2}{128s^2(1+4\sigma^2)\max_l \Sigma_{ll}^2}) \tag{29}$$

Next, we will show that the third inequality in (25) holds. Note that

$$\begin{aligned}
\|(\hat{\mathbf{H}}_{S^c S})^{-1} - (\mathbf{H}_{S^c S})^{-1}\|_\infty &= \|(\mathbf{H}_{SS})^{-1}(\mathbf{H}_{SS} - \hat{\mathbf{H}}_{SS})(\hat{\mathbf{H}}_{SS})^{-1}\|_\infty \\
&\leqslant \sqrt{s}\|(\mathbf{H}_{SS})^{-1}(\mathbf{H}_{SS} - \hat{\mathbf{H}}_{SS})(\hat{\mathbf{H}}_{SS})^{-1}\|_2 \\
&\leqslant \sqrt{s}\|(\mathbf{H}_{SS})^{-1}\|_2\|(\mathbf{H}_{SS} - \hat{\mathbf{H}}_{SS})\|_2\|(\hat{\mathbf{H}}_{SS})^{-1}\|_2
\end{aligned} \tag{30}$$

Note that $\|\mathbf{H}_{SS}\|_2 \geqslant C_{\min}$, thus $\|(\mathbf{H}_{SS})^{-1}\|_2 \leqslant \frac{1}{C_{\min}}$. Similarly, $\|\mathbf{H}_{SS}\|_2 \geqslant \frac{C_{\min}}{2}$ with probability at least $1 - 2\exp(-\frac{cC_{\min}^2 n}{4} + s)$. We also have $\|(\mathbf{H}_{SS} - \hat{\mathbf{H}}_{SS})\|_2 \leqslant \epsilon$ with probability at least $1 - 2\exp(-c\epsilon^2 n + s)$. Taking $\epsilon = \delta\frac{C_{\min}^2}{2\sqrt{s}}$, we get

$$\mathbb{P}(\|(\mathbf{H}_{SS} - \hat{\mathbf{H}}_{SS})\|_2 \geqslant \delta\frac{C_{\min}^2}{2\sqrt{s}}) \leqslant 2\exp(-\frac{c\delta^2 C_{\min}^4 n}{4s} + s) \tag{31}$$

It follows that $\|(\hat{\mathbf{H}}_{SS})^{-1} - (\mathbf{H}_{SS})^{-1}\|_\infty \leqslant \delta$ with probability at least $1 - 2\exp(-\frac{c\delta^2 C_{\min}^4 n}{4s} + s) - 2\exp(-\frac{cC_{\min}^2 n}{4} + s)$. $\qquad\square$

Now we are ready to show that the statement of Lemma 3 holds using the results from Lemma 10. We will rewrite $\hat{\mathbf{H}}_{S^c S}(\hat{\mathbf{H}}_{SS})^{-1}$ as the sum of four different terms:

$$\hat{\mathbf{H}}_{S^c S}(\hat{\mathbf{H}}_{SS})^{-1} = T_1 + T_2 + T_3 + T_4, \tag{32}$$

where

$$T_1 \triangleq \hat{\mathbf{H}}_{S^cS}((\hat{\mathbf{H}}_{SS})^{-1} - (\mathbf{H}_{SS})^{-1})$$
$$T_2 \triangleq (\hat{\mathbf{H}}_{S^cS} - \mathbf{H}_{S^cS})(\mathbf{H}_{SS})^{-1}$$
$$T_3 \triangleq (\hat{\mathbf{H}}_{S^cS} - \mathbf{H}_{S^cS})((\hat{\mathbf{H}}_{SS})^{-1} - (\mathbf{H}_{SS})^{-1}) \tag{33}$$
$$T_4 \triangleq \mathbf{H}_{S^cS}(\mathbf{H}_{SS})^{-1}.$$

Then it follows that $\|\hat{\mathbf{H}}_{S^cS}(\hat{\mathbf{H}}_{SS})^{-1}\|_\infty \leq \|T_1\|_\infty + \|T_2\|_\infty + \|T_3\|_\infty + \|T_4\|_\infty$. Now, we will bound each term separately. First, recall that Assumption 2 ensures that $\|T_4\|_\infty \leq 1 - \alpha$.

**Controlling $T_1$.** We can rewrite $T_1$ as,

$$T_1 = -\mathbf{H}_{S^cS}(\mathbf{H}_{SS})^{-1}(\hat{\mathbf{H}}_{SS} - \mathbf{H}_{SS})(\hat{\mathbf{H}}_{SS})^{-1} \tag{34}$$

then,

$$\begin{aligned}
\|T_1\|_\infty &= \|\mathbf{H}_{S^cS}(\mathbf{H}_{SS})^{-1}(\hat{\mathbf{H}}_{SS} - \mathbf{H}_{SS})(\hat{\mathbf{H}}_{SS})^{-1}\|_\infty \\
&\leq \|\mathbf{H}_{S^cS}(\mathbf{H}_{SS})^{-1}\|_\infty \|(\hat{\mathbf{H}}_{SS} - \mathbf{H}_{SS})\|_\infty \|(\hat{\mathbf{H}}_{SS})^{-1}\|_\infty \\
&\leq (1-\alpha)\|(\hat{\mathbf{H}}_{SS} - \mathbf{H}_{SS})\|_\infty \sqrt{s}\|(\hat{\mathbf{H}}_{SS})^{-1}\|_2 \\
&\leq (1-\alpha)\|(\hat{\mathbf{H}}_{SS} - \mathbf{H}_{SS})\|_\infty \frac{2\sqrt{s}}{C_{\min}} \\
&\leq \frac{\alpha}{6}
\end{aligned} \tag{35}$$

The last inequality holds with probability at least $1 - 2\exp(-\frac{cC_{\min}^2 n}{4} + s) - 4s^2\exp(-\frac{nC_{\min}^2\alpha^2}{18432(1-\alpha)^2 s^3(1+4\sigma^2)\max_l \Sigma_{ll}^2})$ by taking $\delta = \frac{C_{\min}\alpha}{12(1-\alpha)\sqrt{s}}$.

**Controlling $T_2$.** Recall that $T_2 = (\hat{\mathbf{H}}_{S^cS} - \mathbf{H}_{S^cS})(\mathbf{H}_{SS})^{-1}$. Thus,

$$\begin{aligned}
\|T_2\|_\infty &\leq \sqrt{s}\|(\mathbf{H}_{SS})^{-1}\|_2 \|(\hat{\mathbf{H}}_{S^cS} - \mathbf{H}_{S^cS})\|_\infty \\
&\leq \frac{\sqrt{s}}{C_{\min}}\|(\hat{\mathbf{H}}_{S^cS} - \mathbf{H}_{S^cS})\|_\infty \\
&\leq \frac{\alpha}{6}
\end{aligned} \tag{36}$$

The last inequality holds with probability at least $1 - 4(d-s)s\exp(-\frac{nC_{\min}^2\alpha^2}{4608s^3(1+4\sigma^2)\max_l \Sigma_{ll}^2})$ by choosing $\delta = \frac{C_{\min}\alpha}{6\sqrt{s}}$.

**Controlling $T_3$.** Note that,

$$\begin{aligned}
\|T_3\|_\infty &\leq \|(\hat{\mathbf{H}}_{S^cS} - \mathbf{H}_{S^cS})\|_\infty \|((\hat{\mathbf{H}}_{SS})^{-1} - (\mathbf{H}_{SS})^{-1})\|_\infty \\
&\leq \frac{\alpha}{6}
\end{aligned} \tag{37}$$

The last inequality holds with probability at least $1 - 4(d-s)s\exp(-\frac{n\alpha}{768s^2(1+4\sigma^2)\max_l \Sigma_l l^2}) - 2\exp(-\frac{c\alpha C_{\min}^4 n}{24s} + s) - 2\exp(-\frac{cC_{\min}^2 n}{4} + s)$ by choosing $\delta = \sqrt{\frac{\alpha}{6}}$ in the first and third inequality of equation (25). By combining all the above results, we prove Lemma 3. $\square$

# E   Proof of Lemma 4

**Lemma 4** *If Assumptions 1 and 2 hold, $\lambda_n \geq \frac{128\rho k}{\alpha}\frac{\sqrt{\log d}}{n}$ and $n = \Omega(\frac{s^3 \log d}{\tau_1(C_{\min}, \alpha, \sigma, \Sigma, \rho)})$, then the setting of $w$ and $\mathbf{Z}$ from equation (13) satisfies the stationarity condition (8) with probability at least $1 - \mathcal{O}(\frac{1}{d})$, where $\tau_1(C_{\min}, \alpha, \sigma, \Sigma, \rho)$ is a constant independent of $d, s$ or $n$.*

*Proof.* Consider the following optimization problem:

$$\min_w \quad \frac{1}{n}(\mathbf{X}w + \gamma\mathbf{z}^* - \mathbf{y})^\mathsf{T}(\mathbf{X}w + \gamma\mathbf{z}^* - \mathbf{y}) + \lambda_n\|w\|_1 \tag{38}$$

Observe that the above problem is a transformation of optimization problem (6) by fixing $\mathbf{Z} = \mathbf{Z}^*$. With infinite samples (i.e., $n \to \infty, \lambda_n \to 0$), optimization problem (38) is equivalent to the following population version:

$$\min_w \quad \mathbb{E}((Xw + \gamma z^* - y)^\mathsf{T}(Xw + \gamma z^* - y)). \tag{39}$$

Clearly, due to Assumption 1, $w^*$ is the unique optimal solution to (39). Let $\tilde{w}$ be the solution to the optimization problem (38). Notice that after replacing $\mathbf{Z}$ with $\mathbf{Z}^*$ the stationarity condition (8) is same as the stationarity condition for optimization problem (38):

$$\frac{\partial L(w, \mathbf{Z}; \mu, \Lambda)}{\partial w} = \mathbf{0}_{d\times 1} \tag{40}$$

The above simplifies into the following:

$$\frac{2}{n}\mathbf{X}^\mathsf{T}\mathbf{X}\tilde{w} - \frac{2}{n}\mathbf{X}^\mathsf{T}\mathbf{y} + \frac{2\gamma}{n}\mathbf{X}^\mathsf{T}\mathbf{z}^* + \lambda_n\mathbf{g} = \mathbf{0}_{d\times 1}$$

Substituting $\mathbf{y}$ from equation (2), we get:

$$\frac{2}{n}\mathbf{X}^\mathsf{T}\mathbf{X}\Delta - \frac{2}{n}\mathbf{X}^\mathsf{T}\mathbf{e} + \lambda_n\mathbf{g} = \mathbf{0}_{d\times 1}, \tag{41}$$

where $\Delta$ is a short form notation for $\tilde{w} - w^*$. To prove our claim, it suffices to show that $\tilde{w} = (\tilde{w}_S, \mathbf{0}_{d-s\times 1})$ satisfies the stationarity condition (41). This will be true iff $\mathbf{g}_S \in \{-1,1\}^s$ and $\mathbf{g}_{S^c} \in [-1,1]^{d-s}$. In particular, if we start with $w = [w_S, \mathbf{0}_{d-s\times 1}]$ and show that $\|\mathbf{g}_{S^c}\|_\infty < 1$, then our claim holds. To show this, we replace $w$ with $[w_S, \mathbf{0}_{d-s\times 1}]$ and rewrite equation (41) in two parts:

$$\frac{1}{n}\mathbf{X}_S^\mathsf{T}\mathbf{X}_S\Delta_S - \frac{1}{n}\mathbf{X}_S^\mathsf{T}\mathbf{e} + \frac{\lambda_n}{2}\mathbf{g}_S = \mathbf{0}_{s\times 1}, \tag{42}$$

and

$$\frac{1}{n}\mathbf{X}_{S^c}^\mathsf{T}\mathbf{X}_S\Delta_S - \frac{1}{n}\mathbf{X}_{S^c}^\mathsf{T}\mathbf{e} + \frac{\lambda_n}{2}\mathbf{g}_{S^c} = \mathbf{0}_{d-s\times 1}, \tag{43}$$

where $\Delta_S = w_S - w_S^*$. From equation (42):

$$\Delta_S = (\frac{1}{n}\mathbf{X}_S^\mathsf{T}\mathbf{X}_S)^{-1}\frac{1}{n}\mathbf{X}_S^\mathsf{T}\mathbf{e} - (\frac{1}{n}\mathbf{X}_S^\mathsf{T}\mathbf{X}_S)^{-1}\frac{\lambda_n}{2}\mathbf{g}_S$$

By substituting $\Delta_S$ in equation (43), we get:

$$\hat{\mathbf{H}}_{S^cS}(\hat{\mathbf{H}}_{SS}^{-1}\frac{1}{n}\mathbf{X}_S^\mathsf{T}\mathbf{e} - \hat{\mathbf{H}}_{SS}^{-1}\frac{\lambda_n}{2}\mathbf{g}_S) - \frac{1}{n}\mathbf{X}_{S^c}^\mathsf{T}\mathbf{e} + \frac{\lambda_n}{2}\mathbf{g}_{S^c} = \mathbf{0}_{d-s\times 1}$$

By rearranging terms and using the triangle inequality, we get the following:

$$\|\frac{\lambda_n}{2}\mathbf{g}_{S^c}\|_\infty \leqslant \|\hat{\mathbf{H}}_{S^cS}\hat{\mathbf{H}}_{SS}^{-1}\frac{1}{n}\mathbf{X}_S^\mathsf{T}\mathbf{e}\|_\infty + \|\hat{\mathbf{H}}_{S^cS}\hat{\mathbf{H}}_{SS}^{-1}\frac{\lambda_n}{2}\mathbf{g}_S\|_\infty + \|\frac{1}{n}\mathbf{X}_{S^c}^\mathsf{T}\mathbf{e}\|_\infty$$

Using the norm inequality $\|Ab\|_\infty \leqslant \|A\|_\infty\|b\|_\infty$ and noticing that $\|\mathbf{g}_S\|_\infty \leqslant 1$, it follows that:

$$\|\frac{\lambda_n}{2}\mathbf{g}_{S^c}\|_\infty \leqslant \|\hat{\mathbf{H}}_{S^cS}\hat{\mathbf{H}}_{SS}^{-1}\|_\infty(\|\frac{1}{n}\mathbf{X}_S^\mathsf{T}\mathbf{e}\|_\infty + \frac{\lambda_n}{2}) + \|\frac{1}{n}\mathbf{X}_{S^c}^\mathsf{T}\mathbf{e}\|_\infty$$

Furthermore, using Lemma 3, $\|\hat{\mathbf{H}}_{S^cS}\hat{\mathbf{H}}_{SS}^{-1}\|_\infty \leqslant 1 - \frac{\alpha}{2}$ with probability at least $1 - \exp(-\frac{n\tau(C_{\min}, \alpha, \sigma, \Sigma)}{s^2} + \log s)$:

$$\|\mathbf{g}_{S^c}\|_\infty \leqslant (1 - \frac{\alpha}{2})(\|\frac{2}{\lambda_n}\frac{1}{n}\mathbf{X}_S^\mathsf{T}\mathbf{e}\|_\infty + 1) + \|\frac{2}{\lambda_n}\frac{1}{n}\mathbf{X}_{S^c}^\mathsf{T}\mathbf{e}\|_\infty$$

Next, we will need to bound $\|\frac{1}{n}\mathbf{X}_S^\mathsf{T}\mathbf{e}\|_\infty$ and $\|\frac{1}{n}\mathbf{X}_{S^c}^\mathsf{T}\mathbf{e}\|_\infty$ which we do in the following lemma:

**Lemma 11.** *Let* $\lambda_n \geqslant \frac{128\rho k}{\alpha}\frac{\sqrt{\log d}}{n}$ *and* $n \geqslant \frac{\log d}{(1-\frac{\alpha}{2})^2}$. *Then the following holds true:*

$$\mathbb{P}(\|\frac{2}{\lambda_n}\frac{1}{n}\mathbf{X}_S^\mathsf{T}\mathbf{e}\|_\infty \geqslant \frac{\alpha}{8-4\alpha}) \leqslant \mathcal{O}(\frac{1}{d}), \quad \mathbb{P}(\|\frac{2}{\lambda_n}\frac{1}{n}\mathbf{X}_{S^c}^\mathsf{T}\mathbf{e}\|_\infty \geqslant \frac{\alpha}{8}) \leqslant \mathcal{O}(\frac{1}{d})$$

Using results from Lemma 11, we show that $\|\mathbf{g}_{S^c}\|_\infty \leqslant 1 - \frac{\alpha}{4}$ with probability at least $1 - \mathcal{O}(\frac{1}{d})$. This ensures that $\tilde{w} = (\tilde{w}_S, \mathbf{0}_{d-s\times 1})$ indeed satisfies the stationarity condition (8). $\qquad\square$

# F Proof of Lemma 11

**Lemma 11** *Let $\lambda_n \geqslant \frac{128\rho k}{\alpha} \frac{\sqrt{\log d}}{n}$ and $n \geqslant \frac{\log d}{(1-\frac{\alpha}{2})^2}$. Then the following holds true:*

$$\mathbb{P}(\|\frac{2}{\lambda_n} \frac{1}{n} \mathbf{X}_S^\intercal \mathbf{e}\|_\infty \geqslant \frac{\alpha}{8-4\alpha}) \leqslant \mathcal{O}(\frac{1}{d})$$

$$\mathbb{P}(\|\frac{2}{\lambda_n} \frac{1}{n} \mathbf{X}_{S^c}^\intercal \mathbf{e}\|_\infty \geqslant \frac{\alpha}{8}) \leqslant \mathcal{O}(\frac{1}{d}) \tag{44}$$

*Proof.* We will start with $\frac{1}{n}\mathbf{X}_S^\intercal \mathbf{e}$. We take the $i$-th entry of $\frac{1}{n}\mathbf{X}_S^\intercal \mathbf{e}$ for some $i \in S$. Note that

$$|\frac{1}{n}\mathbf{X}_{i.}^\intercal \mathbf{e}| = |\frac{1}{n} \sum_{j=1}^n \mathbf{X}_{ji}\mathbf{e}_j| \tag{45}$$

Recall that $\mathbf{X}_{ji}$ is a sub-Gaussian random variable with parameter $\rho^2$ and $\mathbf{e}_j$ is a sub-Gaussian random variable with parameter $\sigma_e^2$. Then, $\frac{\mathbf{X}_{ji}}{\rho} \frac{\mathbf{e}_j}{\sigma_e}$ is a sub-exponential random variable with parameters $(4\sqrt{2}, 2)$. Using the concentration bounds for the sum of independent sub-exponential random variables (Wainwright, 2019), we can write:

$$\mathbb{P}(|\frac{1}{n} \sum_{j=1}^n \frac{\mathbf{X}_{ji}}{\rho} \frac{\mathbf{e}_j}{\sigma_e}| \geqslant t) \leqslant 2\exp(-\frac{nt^2}{64}), \ 0 \leqslant t \leqslant 8 \tag{46}$$

Taking a union bound across $i \in S$:

$$\mathbb{P}(\exists i \in S \mid |\frac{1}{n} \sum_{j=1}^n \frac{\mathbf{X}_{ji}}{\rho} \frac{\mathbf{e}_j}{\sigma_e}| \geqslant t) \leqslant 2s\exp(-\frac{nt^2}{64})$$

$$0 \leqslant t \leqslant 8 \tag{47}$$

Taking $t = \frac{\lambda_n t}{2\rho\sigma_e}$, we get:

$$\mathbb{P}(\exists i \in S \mid |\frac{2}{\lambda_n} \frac{1}{n} \sum_{j=1}^n \mathbf{X}_{ji}\mathbf{e}_j| \geqslant t) \leqslant 2s\exp(-\frac{n\lambda_n^2 t^2}{256\rho^2\sigma_e^2})$$

$$0 \leqslant t \leqslant 16\frac{\rho\sigma_e}{\lambda_n} \tag{48}$$

It follows that $\|\frac{2}{\lambda} \frac{1}{n} \mathbf{X}_S^\intercal \mathbf{e}\|_\infty \leqslant t$ with probability at least $1 - 2s\exp(-\frac{n\lambda_n^2 t^2}{256\rho^2\sigma_e^2})$.

Using a similar argument, we can show that $\|\frac{2}{\lambda} \frac{1}{n} \mathbf{X}_{S^c}^\intercal \mathbf{e}\|_\infty \leqslant t$ with probability at least $1 - 2(d-s)\exp(-\frac{n\lambda_n^2 t^2}{256\rho^2\sigma_e^2})$. Taking $t = \frac{\alpha}{8-4\alpha}$ and $\frac{\alpha}{8}$ in the first and second inequality of Lemma 11 and choosing the provided setting of $\lambda_n$ and $n$ completes our proof. $\qquad\square$

# G Proof of Lemma 8

**Lemma 8** *If Assumptions 1 and 2 hold, $\lambda_n \geqslant \frac{128\rho k\sqrt{\log d}}{\alpha n}$ and $n = \Omega(\frac{s^3 \log d}{\tau_2(C_{\min},\rho,k)})$, then $\|\Delta_S\|_2 \leqslant \frac{2\lambda_n\sqrt{s}}{C_{\min}}$ with probability at least $1 - \mathcal{O}(\frac{1}{d})$ where $\tau_2(C_{\min},\rho,k)$ is a constant independent of $s, d$ or $n$.*

*Proof.* Using results from Lemma 4, we can write:

$$\|\Delta_S\|_2 \leqslant \|\hat{\mathbf{H}}_{SS}^{-1} \frac{1}{n}\mathbf{X}_S^\intercal \mathbf{e}\|_2 + \|\hat{\mathbf{H}}_{SS}^{-1} \frac{\lambda_n}{2}\mathbf{g}_S\|_2$$

Using the norm inequality $\|Ab\|_2 \leqslant \|A\|_2\|b\|_2$ and noticing that $\|\mathbf{g}_S\|_2 \leqslant \sqrt{s}$, we can rewrite the above equation as:

$$\|\Delta_S\|_2 \leqslant \|\hat{\mathbf{H}}_{SS}^{-1}\|_2(\|\frac{1}{n}\mathbf{X}_S^\intercal \mathbf{e}\|_2 + \frac{\lambda_n}{2}\sqrt{s})$$

Using Assumption 1 and results from Lemma 2 and substituting $\|\hat{\mathbf{H}}_{SS}^{-1}\|_2 \leqslant \frac{2}{C_{\min}}$ in the above inequality, we get:

$$\|\Delta_S\|_2 \leqslant \frac{2}{C_{\min}}(\|\frac{1}{n}\mathbf{X}_S^\mathsf{T}\mathbf{e}\|_2 + \frac{\lambda_n}{2}\sqrt{s})$$

We present the next lemma to bound the term $\|\frac{1}{n}\mathbf{X}_S^\mathsf{T}\mathbf{e}\|_2$.

**Lemma 12.** If $\lambda_n \geqslant \frac{128\rho k}{\alpha}\frac{\sqrt{\log d}}{n}$ and $n = \Omega(\frac{s^3 \log d}{\tau_2(C_{\min},\rho,k)})$, then $\|\frac{1}{n}\mathbf{X}_S^\mathsf{T}\mathbf{e}\|_2 \leqslant \sqrt{s}\frac{\lambda_n}{2}$ with probability at least $1 - \mathcal{O}(\frac{1}{d})$.

We take $t = \frac{\lambda_n}{2}$ in the above lemma and get $\|\Delta\|_2 \leqslant \frac{2\lambda_n\sqrt{s}}{C_{\min}}$ with probability at least $1 - \mathcal{O}(\frac{1}{d})$. $\quad\square$

# H   Proof of Lemma 12

**Lemma 12**   If $\lambda_n \geqslant \frac{128\rho k}{\alpha}\frac{\sqrt{\log d}}{n}$ and $n = \Omega(\frac{s^3 \log d}{\tau_2(C_{\min},\rho,k)})$, then $\|\frac{1}{n}\mathbf{X}_S^\mathsf{T}\mathbf{e}\|_2 \leqslant \sqrt{s}\frac{\lambda_n}{2}$ with probability at least $1 - \mathcal{O}(\frac{1}{d})$.

*Proof.* We take the $i$-th entry of $\frac{1}{n}\mathbf{X}_S^\mathsf{T}\mathbf{e}$ for some $i \in S$. Note that

$$|\frac{1}{n}\mathbf{X}_{i.}^\mathsf{T}\mathbf{e}| = |\frac{1}{n}\sum_{j=1}^n \mathbf{X}_{ji}\mathbf{e}_j| \tag{49}$$

Recall that $\mathbf{X}_{ji}$ is a sub-Gaussian random variable with parameter $\rho$ and $\mathbf{e}_j$ is a sub-Gaussian random variable with parameter $\sigma_e^2$). Then, $\frac{\mathbf{X}_{ji}}{\rho}\frac{\mathbf{e}_j}{\sigma_e}$ is a sub-exponential random variable with parameters $(4\sqrt{2}, 2)$. Using the concentration bounds for the sum of independent sub-exponential random variables (Wainwright, 2019), we can write:

$$\mathbb{P}(|\frac{1}{n}\sum_{j=1}^n \frac{\mathbf{X}_{ji}}{\rho}\frac{\mathbf{e}_j}{\sigma_e}| \geqslant t) \leqslant 2\exp(-\frac{nt^2}{64}), \ 0 \leqslant t \leqslant 8 \tag{50}$$

Taking a union bound across $i \in S$, we get

$$\mathbb{P}(\exists i \in S \mid |\frac{1}{n}\sum_{j=1}^n \frac{\mathbf{X}_{ji}}{\rho}\frac{\mathbf{e}_j}{\sigma_e}| \geqslant t) \leqslant 2s\exp(-\frac{nt^2}{64}),$$
$$0 \leqslant t \leqslant 8 \tag{51}$$

It follows that $\|\frac{1}{n}\mathbf{X}_S^\mathsf{T}\mathbf{e}\|_2 \leqslant \sqrt{s}t$ with probability at least $1 - 2s\exp(-\frac{nt^2}{64\rho^2\sigma_e^2})$ for some $0 \leqslant t \leqslant 8\rho\sigma_e$. $\quad\square$

# I   Proof of Corollary 1

**Corollary 1**   If Assumptions 1 and 2 hold, $\lambda_n \geqslant \frac{128\rho k}{\alpha}\frac{\sqrt{\log d}}{n}$ and $n = \Omega(\frac{s^3 \log d}{\tau_1(C_{\min},\alpha,\sigma,\Sigma,\rho)})$, then the following statements are true with probability at least $1 - \mathcal{O}(\frac{1}{n})$:

1. *The solution* $\mathbf{Z}$ *correctly recovers hidden attribute for each sample, i.e.,* $\mathbf{Z} = \mathbf{Z}^* = \zeta^*\zeta^{*\mathsf{T}}$.

2. *The support of recovered regression parameter* $\tilde{w}$ *matches exactly with the support of* $w^*$.

3. *If* $\min_{i \in S}|w_i^*| \geqslant \frac{4\lambda_n\sqrt{s}}{C_{\min}}$ *then for all* $i \in [d]$, $\tilde{w}_i$ *and* $w_i^*$ *match up to their sign.*

*Proof.* Since $\mathbf{Z} = \mathbf{Z}^*$, the hidden attributes of each sample can be read by simply looking at the first row or column of $\mathbf{Z}$ and skipping the first entry. The supports of $\tilde{w}$ and $w^*$ match exactly through construction (and subsequent proofs). Observe that $\|\Delta\|_\infty \leqslant \|\Delta\|_2 \leqslant \frac{2\lambda_n\sqrt{s}}{C_{\min}}$. Thus, it follows that if $\min_{i \in S}|w_i^*| \geqslant \frac{4\lambda_n\sqrt{s}}{C_{\min}}$ then for all $i \in [d]$, $\tilde{w}_i$ and $w_i^*$ will have the same sign. $\quad\square$

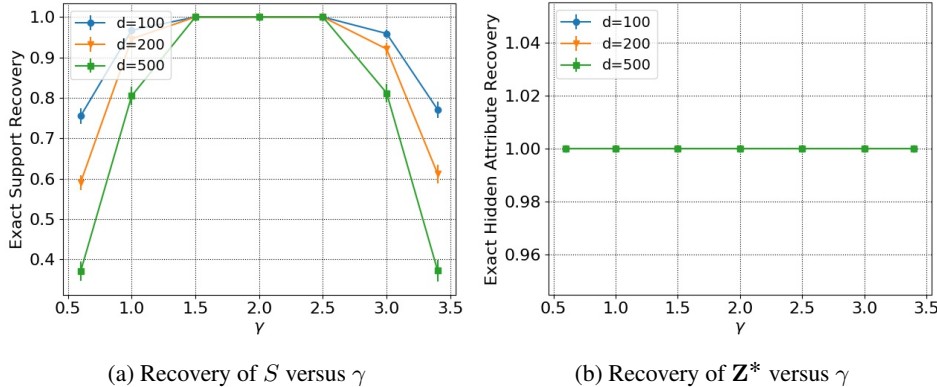

(a) Recovery of $S$ versus $\gamma$          (b) Recovery of $\mathbf{Z}^*$ versus $\gamma$

Figure 3: Left: Exact support recovery of $w^*$ across 30 runs. Right: Exact hidden attribute recovery of $\mathbf{Z}_*$ across 30 runs. The true value of $\gamma$ is 2.

## J Quality of Solution with bias parameter $\gamma$

Our method requires a known value of bias parameter $\gamma$ in our analysis. However, in practice, we observe that even a rough estimate (up to $\pm 25\%$) works pretty well. We conducted computational experiments with a range of values of $\gamma$ and the reported results are averaged across 30 independent runs. The performance measures used here are the same as in Section 5 (See Appendix M for details). Figure 3a shows the quality of support recovery for different values of $\gamma$ and Figure 3b shows the quality of recovering the hidden attributes for different values of $\gamma$. Note how both the curves show $100\%$ correct recovery for a wide range of $\gamma$. These experiments show that prior knowledge of the exact value of $\gamma$ is not necessary for our method.

## K Alternate Optimization Algorithm for Solving Optimization Problem (6)

We use the following alternate optimization algorithm to solve optimization problem (6) in our computational experiments.

---

**Input:** Data samples $(\mathbf{X}, \mathbf{y})$, amount of bias $\gamma$
**Output:** $\tilde{w}, \mathbf{Z}$
$\mathbf{Z}_0 \leftarrow \mathbf{I}_{n+1 \times n+1}$
$\mathbf{z}_0 \leftarrow \mathbf{Z}_0(2 : n+1, 1)$
**for** $t = 1, 2, \cdots$ until $\mathbf{Z}_{t-1} = \mathbf{Z}_t$ **do**
   $\tilde{w}_t \leftarrow \arg\min_w \frac{1}{n}(\mathbf{X}w + \gamma\mathbf{z}_{t-1} - \mathbf{y})^\intercal(\mathbf{X}w + \gamma\mathbf{z}_{t-1} - \mathbf{y}) + \lambda_n\|w\|_1$
   $\mathbf{M}(\tilde{w}_t) \leftarrow \begin{bmatrix} \frac{1}{n}\|\mathbf{X}\tilde{w}_t - \mathbf{y}\|_2^2 & \frac{\gamma}{n}(\mathbf{X}\tilde{w}_t - \mathbf{y})^\intercal \\ \frac{\gamma}{n}(\mathbf{X}\tilde{w}_t - \mathbf{y})^\intercal & \frac{\gamma^2}{n}\mathbf{I}_{n \times n} \end{bmatrix}$
   $\mathbf{Z}_t \leftarrow \arg\min_{\mathbf{Z}} \operatorname{trace}(\mathbf{M}(\tilde{w}_t)\mathbf{Z})$, such that $\operatorname{diag}(\mathbf{Z}) = \mathbf{1}, \mathbf{Z} \succeq \mathbf{0}_{n+1 \times n+1}$
   $\mathbf{z}_t \leftarrow \mathbf{Z}_t(2 : n+1, 1)$
**end for**
$\tilde{w} \leftarrow \tilde{w}_t, \;\; \mathbf{Z} \leftarrow \mathbf{Z}_t$

**Algorithm 1:** Alternate Optimization Algorithm for our problem

---

Recall from equation (5) that

$$\mathbf{Z} \triangleq \begin{bmatrix} 1 & \mathbf{z}^\intercal \\ \mathbf{z} & \mathbf{z}\mathbf{z}^\intercal \end{bmatrix}. \tag{52}$$

Thus, we can read $\mathbf{z}$ from $\mathbf{Z}$ by considering its first column and skipping the first entry. We denote this as $\mathbf{z} = \mathbf{Z}(2 : n+1, 1)$. We use a similar notation in Algorithm 1 to assign values to vectors $\mathbf{z}_0$ and $\mathbf{z}_t$ from matrices $\mathbf{Z}_0$ and $\mathbf{Z}_t$ respectively.

We will show that if Algorithm 1 converges then it converges to the optimal solution of optimization problem (6). To do this, consider

$$f_1(w, \mathbf{Z}) = \frac{1}{n}(\mathbf{X}w + \gamma \mathbf{z}_{t-1} - \mathbf{y})^\mathsf{T}(\mathbf{X}w + \gamma \mathbf{z}_{t-1} - \mathbf{y})$$
$$f_2(w) = \lambda_n \|w\|_1 .$$

(53)

Note that $f_2(w)$ is not differentiable. Let $g(\mathbf{Z}) \triangleq -\mathrm{eig}_{\min}(\mathbf{Z})$ and $h_i(\mathbf{Z}) \triangleq \mathbf{Z}_{ii} - 1, \forall i \in [n+1]$. Observe that $g(\mathbf{Z}) \leqslant 0$ and $h_i(\mathbf{Z}) = 0, \forall i \in [n+1]$ denote the constraints $\mathbf{Z} \succeq \mathbf{0}_{n+1 \times n+1}$ and $\mathrm{diag}(\mathbf{Z}) = \mathbf{1}$ respectively. We define $\frac{\partial f_2(w)}{\partial w}$ as the sub-differential set for $f_2(w)$ and $f_2'(w) \in \frac{\partial f_2(w)}{\partial w}$ is an element of the sub-differential set $\frac{\partial f_2(w)}{\partial w}$. Observe that $f_1(w, \mathbf{Z}) + f_2(w)$, $g(\mathbf{Z})$ and $h_i(\mathbf{Z})$ are convex with respect to $w$ and $\mathbf{Z}$ separately but they are not jointly convex. Consider the following optimization problem:

$$\tilde{w}, \mathbf{Z}^* = \begin{array}{ll} \arg\min_{w, \mathbf{Z}} & f_1(w, \mathbf{Z}) + f_2(w) \\ \text{such that} & g(\mathbf{Z}) \leqslant 0 \\ & h_i(\mathbf{Z}) = 0 \qquad \forall i \in [n+1] \end{array}$$

(54)

We have already shown that the solution $\tilde{w}, \mathbf{Z}^*$ is the unique solution to (54). We propose the following alternate optimization algorithm to solve this problem:

**Output:** $w, \mathbf{Z}$
$\mathbf{Z}_0 \leftarrow \mathbf{I}_{n+1 \times n+1}$
**for** $t = 1, 2 \cdots$ until $\mathbf{Z}_{t-1} = \mathbf{Z}_t$ **do**

$$w_t \leftarrow \arg\min_w f_1(w, \mathbf{Z}_{t-1}) + f_2(w)$$

(55)

$$\mathbf{Z}_t \leftarrow \begin{array}{ll} \arg\min_{\mathbf{Z}} & f_1(w_t, \mathbf{Z}) \\ \text{such that} & g(\mathbf{Z}) \leqslant 0 \\ & h_i(\mathbf{Z}) = 0 \quad \forall i \in [n+1] \end{array}$$

(56)

**end for**
$w \leftarrow w_t, \quad \mathbf{Z} \leftarrow \mathbf{Z}_t$

**Algorithm 2:** Alternate Optimization Algorithm

We will prove the following proposition:

**Proposition 1.** *If Algorithm 2 converges, then $w = \tilde{w}$ and $\mathbf{Z} = \mathbf{Z}^*$.*

*Proof.* We start by writing the KKT conditions for optimization problem (54).

1. Stationarity conditions: $\frac{\partial f_1(\tilde{w}, \mathbf{Z}^*)}{\partial w} + f_2'(\tilde{w}) = 0$ and $\frac{\partial f_1(\tilde{w}, \mathbf{Z}^*)}{\partial \mathbf{Z}} + r\frac{\partial g(\mathbf{Z}^*)}{\partial \mathbf{Z}} + \sum_{i=1}^{n+1} s_i \frac{\partial h_i(\mathbf{Z}^*)}{\partial \mathbf{Z}} = 0$.

2. Complementary slackness condition: $rg(\mathbf{Z}^*) = 0$.

3. Primal feasibility condition: $g(\mathbf{Z}^*) \leqslant 0$ and $h_i(\mathbf{Z}^*) = 0, \forall i \in [n+1]$.

4. Dual feasibility condition: $r \geqslant 0$.

Any optimal solution to optimization problem (54) must satisfy the above KKT conditions. Next, we write the KKT conditions for (55) at convergence, i.e., at $\mathbf{Z}_t = \mathbf{Z}_{t-1}$:

1. Stationarity condition: $\frac{\partial f(w_t, \mathbf{Z}_t)}{\partial w} + f_2'(w_t) = 0$

Similarly, we write the KKT conditions for (56) at convergence, i.e., at $\mathbf{Z}_t = \mathbf{Z}_{t-1}$:

1. Stationarity conditions: $\frac{\partial f_1(w_t, \mathbf{Z}_t)}{\partial \mathbf{Z}} + t\frac{\partial g(\mathbf{Z}_t)}{\partial \mathbf{Z}} + \sum_{i=1}^{n+1} u_i \frac{\partial h_i(\mathbf{Z}_t)}{\partial \mathbf{Z}} = 0$.

2. Complementary slackness condition: $tg(\mathbf{Z}_t) = 0$.

3. Primal feasibility condition: $g(\mathbf{Z}_t) \leqslant 0$ and $h_i(\mathbf{Z}_t) = 0, \forall i \in [n+1]$.

4. Dual feasibility condition: $t \geqslant 0$.

Combining the KKT conditions at $w_t, \mathbf{Z}_t$ for (55) and (56) and taking $r = t$ and $s_i = u_i, \forall i \in [n+1]$, we see that all KKT conditions of (54) are satisfied by $w_t, \mathbf{Z}_t$. Since the solution to (54) is unique, it follows that $w = \tilde{w}$ and $\mathbf{Z} = \mathbf{Z}^*$. $\qquad\square$

## L  Our Assumptions Hold for Finite Samples

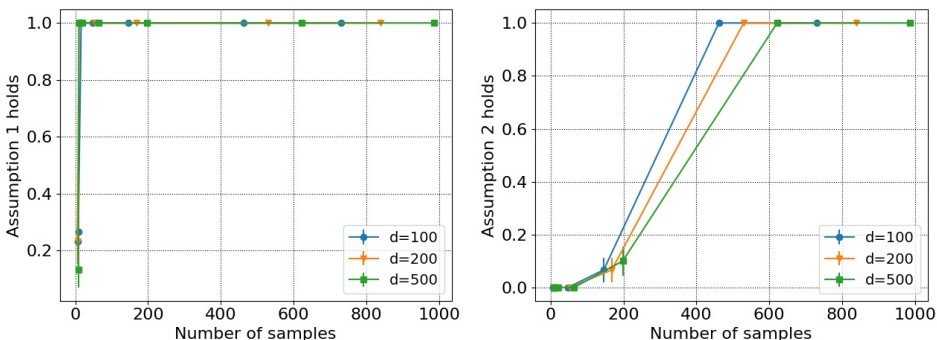

(a) Positive Definiteness against number of samples

(b) Mutual Incoherence against number of samples

Figure 4: Left: Positive Definiteness Assumption 1 with varying number of samples for $d = 100, 200$ and $500$, Right: Mutual Incoherence Assumption 2 with varying number of samples for $d = 100, 200$ and $d = 500$.

Figure 4 shows how our assumptions hold (averaged across 30 independent runs) in the finite sample regime with varying number of samples when $X$ is drawn from a standard normal distribution. We notice that for a fixed $s$, Assumption 1 is easier to hold (i.e., $n = \Omega(s + \log d)$) than Assumption 2 (i.e., $n = \Omega(s^3 \log d)$). Eventually, both assumptions hold as the number of samples increases.

## M  Details of Experimental Validation

In this section, we validate our theoretical results by conducting computational experiments on synthetic data. We will show that for a fixed $s$, we need $n = 10^\beta \log d$ samples for recovering the exact support of $w^*$ and exact hidden attributes $\mathbf{Z}^*$, where $\beta \equiv \beta(s, C_{\min}, \alpha, \sigma, \Sigma, \rho, \gamma, k)$ is a control parameter which is independent of $d$.

**Data Generation.** For $d = 100, 200$ and $500$, we draw $\mathbf{X} \in \mathbb{R}^{n \times d}$ from a standard Gaussian distribution by varying $n$ as $10^\beta \log d$ for a control parameter $\beta$. The $s = 10$ non-zero entries of true parameter $w^* \in \mathbb{R}^d$ are chosen uniformly at random between $[-1, 1]$. Every non-zero entry in $w^*$ is changed to at least $0.75$ to make sure that it is not too close to 0. The independent noise $\mathbf{e} \in \mathbb{R}^n$ is drawn from a zero mean Gaussian distribution with standard deviation $\frac{k}{\sqrt{\log n}}$ for $k = 0.15$. The estimate of the bias $\gamma \in \mathbb{R}_{>0}$ is kept at 2. Regarding the hidden attribute $\mathbf{z}^* \in \{-1, 1\}^n$, we set $\frac{n}{2}$ entries as $+1$ and the rest as $-1$. The response $\mathbf{y} \in \mathbb{R}^n$ is generated according to (1). This process is repeated 30 times and the reported results are averaged across these 30 independent runs.

**Choice of Regularizer and Solution.** According to Theorem 1, the regularizer $\lambda_n$ is chosen to be equal to $\frac{128\rho k}{\alpha} \frac{\sqrt{\log d}}{n}$. We solve optimization problem (6) by using an alternate optimization algorithm that converges to the optimal solution (See Appendix K for details).

**Measure of Performance.** The performance is measured by comparing the recovered solutions $\tilde{w}$ and $\mathbf{Z}$ with the true parameters $w^*$ and $\mathbf{Z}^*$. The quality of $\tilde{w}$ is measured by comparing its support to the support $S$ of the true parameter $w^*$ by computing the Jaccard index $J(S, \hat{S})$, where $\hat{S}$ is the support of $\tilde{w}$, i.e., $\hat{S} = \{i | \tilde{w}_i \neq 0, i \in [d]\}$. The average of $J(S, \hat{S})$ across 30 independent runs is plotted against the number of samples $n$ (See Figure 2a, 2b). Similarly, the quality of $\mathbf{Z}$ is measured by the indicator variable $I(\mathbf{Z}, \mathbf{Z}^*)$. The average of $I(\mathbf{Z}, \mathbf{Z}^*)$ across 30 independent runs is plotted against the number of samples $n$ (See Figure 2c, 2d). The Jaccard index $J(S, \hat{S})$ and indicator variable $I(\mathbf{Z}, \mathbf{Z}^*)$ are defined as follows:

$$J(S, \hat{S}) \triangleq \frac{|S \cap \hat{S}|}{|S \cup \hat{S}|}, \quad I(\mathbf{Z}, \mathbf{Z}^*) \triangleq \begin{cases} 0, \text{ if } \mathbf{Z} \neq \mathbf{Z}^* \\ 1 \text{ if } \mathbf{Z} = \mathbf{Z}^* \end{cases}$$

**Observation.** Figure 2a shows the Jaccard index of support recovery with varying number of samples. We see that our method recovers the true support for all three values of $d$ as we increase number of samples. Also, notice how all three curves line up perfectly in Figure 2b when we plot the support recovery with respect to the control parameter $\beta = \log \frac{n}{\log d}$. This validates our theoretical results. Similarly, Figure 2c shows exact recovery of the hidden attribute with varying number of samples. We again see that as the number of samples increase, our recovered hidden attributes are 100% correct. Again, the three different curves for different values of $d$ line up nicely when plotted against $\beta$. Interestingly, a small percentage of our experiments recover the hidden attributes exactly for small number of samples ($< 20$). We believe that this can be ascribed to $\mathbf{Z}^*$ having small dimensions and thus becoming relatively easier to recover. On a more practical point of view, once hidden attributes are identified for each sample point, the associated bias (for and against) can be duly removed from the model.

## N   Optimization Problem (6) is Non-Convex

Before we begin the proof of non-convexity of (6), we note that optimization (6) is stated for a fixed $\mathbf{X}$. However, since the entries in $\mathbf{X}$ are drawn from a sub-Gaussian distribution, matrix $\mathbf{X}$ can potentially realize any real matrix in $\mathbb{R}^{n \times d}$. In particular, we are interested in a problem where $\exists i, k \in [d]$ such that $\sum_{l=1}^{n} \mathbf{X}_{li}^2 - \mathbf{X}_{ki}$ is non-zero. Since $\mathbf{X}$ can be any real matrix in $\mathbb{R}^{n \times d}$, this is not a strong assumption. With this assumption in mind, we present the following lemma.

**Lemma 13.** *The optimization problem* (6) *defined on a convex set $C$, is non-convex.*

*Proof.* As defined in (6), we define the domain for optimization problem on a convex set $C = \{(w, \mathbf{Z}) \mid w \in \mathbb{R}^d, \text{diag}(\mathbf{Z}) = 1, \mathbf{Z} \geq \mathbf{0}_{n+1 \times n+1}\}$. It should be noted that $C$ is a convex set and we will show that the non-convexity of the problem comes from the objective function. We are solving the following optimization problem:

$$\min_{(w, \mathbf{Z}) \in C} \quad \langle \mathbf{M}(w), \mathbf{Z} \rangle + \lambda_n \|w\|_1, \tag{57}$$

It suffices to show that $f(w, \mathbf{Z}) = \langle \mathbf{M}(w), \mathbf{Z} \rangle$ is non-convex function. To that end, we will construct a setting of $(w, \mathbf{Z}) \in C$ and $(\bar{w}, \bar{\mathbf{Z}}) \in C$ such that the first order condition for convexity fails to hold, i.e,

$$f(w, \mathbf{Z}) - f(\bar{w}, \bar{\mathbf{Z}}) < \left\langle \frac{\partial f(\bar{w}, \bar{\mathbf{Z}})}{\partial w}, (w - \bar{w}) \right\rangle + \left\langle \frac{\partial f(\bar{w}, \bar{\mathbf{Z}})}{\partial \mathbf{Z}}, \mathbf{Z} - \bar{\mathbf{Z}} \right\rangle. \tag{58}$$

First notice that,

$$\frac{\partial f(\bar{w}, \bar{\mathbf{Z}})}{\partial w} = \sum_{ij} \bar{\mathbf{Z}}_{ij} \frac{\partial \mathbf{M}_{ij}(\bar{w})}{\partial w}, \quad \frac{\partial f(\bar{w}, \bar{\mathbf{Z}})}{\partial \mathbf{Z}} = \mathbf{M}(\bar{w})$$

Recall from equation (5) that,

$$l(w) \triangleq \frac{1}{n}(\mathbf{X}w - \mathbf{y})^\intercal(\mathbf{X}w - \mathbf{y}), \quad \mathbf{M}(w) \triangleq \begin{bmatrix} l(w) & \frac{\gamma}{n}(\mathbf{X}w - \mathbf{y})^\intercal \\ \frac{\gamma}{n}(\mathbf{X}w - \mathbf{y}) & \frac{\gamma^2}{n}\mathbf{I}_{n \times n} \end{bmatrix}, \tag{59}$$

Then $\frac{\partial f(\bar{w},\mathbf{Z})}{\partial w}$ can be simplified as:

$$\frac{\partial f(\bar{w},\bar{\mathbf{Z}})}{\partial w} = \frac{2}{n}(\mathbf{X}^\mathsf{T}\mathbf{X}\bar{w} - \mathbf{X}^\mathsf{T}\mathbf{y} + \mathbf{X}^\mathsf{T}\bar{\mathbf{z}}), \tag{60}$$

where $\bar{\mathbf{z}} \in \mathbb{R}^n$ denotes the first column of $\bar{\mathbf{Z}}$ after skipping the first entry.

We provide the following construction for $(w,\mathbf{Z}) \in C$ and $(\bar{w},\bar{\mathbf{Z}}) \in C$. We take $w \in \{0,\beta\}^d$ such that $w_k = 0, \forall k \neq i$ and $w_i = \beta$ where $\beta \in \mathbb{R}$. Similarly, $\bar{w} \in \{0,\beta\}^d$ such that $\bar{w}_k = 0, \forall k \neq i$ and $\bar{w}_i = -\beta$. Since $w \in \mathbb{R}^d$, such a setting exists for a non-zero $\beta$. Furthermore, we take $\mathbf{Z} = \mathbf{I}_{n+1 \times n+1}$ and $\bar{\mathbf{Z}} \in \{0,1\}^{n+1 \times n+1}$ such that $\bar{\mathbf{Z}}_{ii} = 1, \forall i \in [n+1]$ and $\bar{\mathbf{Z}}_{1(k+1)} = 1, \bar{\mathbf{Z}}_{(k+1)1} = 1$. Now, we can compute the following quantities:

$$\langle \mathbf{M}(w), \mathbf{Z} \rangle = l(w) + \gamma^2 = \frac{1}{n}\sum_{l=1}^n (\mathbf{X}_{li}w_i - y_l)^2 + \gamma^2$$

$$\langle \mathbf{M}(\bar{w}), \bar{\mathbf{Z}} \rangle = l(\bar{w}) + \gamma^2 - \frac{2\gamma}{n}(\mathbf{X}_{ki}\bar{w}_i - y_k) = \frac{1}{n}\sum_{l=1}^n (\mathbf{X}_{li}\bar{w}_i - y_l)^2 + \gamma^2 + \frac{2\gamma}{n}(\mathbf{X}_{ki}\bar{w}_i - y_k)$$

$$\langle \mathbf{M}(\bar{w}), \mathbf{Z} - \bar{\mathbf{Z}} \rangle = -\frac{2\gamma}{n}(\mathbf{X}_{ki}\bar{w}_i - y_k)$$

$$\langle \frac{\partial f(\bar{w},\bar{\mathbf{Z}})}{\partial w}, w - \bar{w} \rangle = \frac{2}{n}((w_i\bar{w}_i - \bar{w}_i^2)\sum_{l=1}^n \mathbf{X}_{li}^2 + (-w_i + \bar{w}_i)\sum_{l=1}^n \mathbf{X}_{li}\mathbf{y}_l + (w_i - \bar{w}_i)\mathbf{X}_{ki})$$

$$\tag{61}$$

Substituting $w_i = \beta$ and $\bar{w}_i = -\beta$, we get

$$l(w) - l(\bar{w}) = -\frac{4\beta}{n}\sum_{l=1}^n \mathbf{X}_{li}\mathbf{y}_l$$

$$\langle \frac{\partial f(\bar{w},\bar{\mathbf{Z}})}{\partial w}, w - \bar{w} \rangle = -\frac{4\beta}{n}\sum_{l=1}^n \mathbf{X}_{li}^2 - \frac{4\beta}{n}\sum_{l=1}^n \mathbf{X}_{li}\mathbf{y}_l + \frac{4\beta}{n}\mathbf{X}_{ki}$$

$$\tag{62}$$

Clearly,

$$\langle \mathbf{M}(w), \mathbf{Z} \rangle - \langle \mathbf{M}(\bar{w}), \bar{\mathbf{Z}} \rangle = -\frac{4\beta}{n}\sum_{l=1}^n \mathbf{X}_{li}\mathbf{y}_l - \frac{2\gamma}{n}(\mathbf{X}_{ki}\bar{w}_i - y_k)$$

$$\langle \mathbf{M}(\bar{w}), \mathbf{Z} - \bar{\mathbf{Z}} \rangle + \langle \frac{\partial f(\bar{w},\bar{\mathbf{Z}})}{\partial w}, w - \bar{w} \rangle = -\frac{2\gamma}{n}(\mathbf{X}_{ki}\bar{w}_i - y_k) - \frac{4\beta}{n}\sum_{l=1}^n \mathbf{X}_{li}^2 - \frac{4\beta}{n}\sum_{l=1}^n \mathbf{X}_{li}\mathbf{y}_l + \frac{4\beta}{n}\mathbf{X}_{ki}$$

$$\tag{63}$$

It follows that

$$\langle \mathbf{M}(w), \mathbf{Z} \rangle - \langle \mathbf{M}(\bar{w}), \bar{\mathbf{Z}} \rangle - \langle \mathbf{M}(\bar{w}), \mathbf{Z} - \bar{\mathbf{Z}} \rangle - \langle \frac{\partial f(\bar{w},\bar{\mathbf{Z}})}{\partial w}, w - \bar{w} \rangle = \beta(\frac{4}{n}\sum_{l=1}^n \mathbf{X}_{li}^2 - \frac{4}{n}\mathbf{X}_{ki})$$

$$\tag{64}$$

As $\sum_{l=1}^n \mathbf{X}_{li}^2 - \mathbf{X}_{ki}$ is assumed to be non-zero, it is easy to see that LHS of equation (64) can be made greater than or less than 0 by simply choosing appropriate $\beta \in \mathbb{R}$. Thus, optimization problem (6) is non-convex. $\qquad\square$

## O   Real World Experiment

We show applicability of our method by conducting experiments on Communities and Crime Data Set (Redmond, 2002) and Student Performance Data Set (Cortez, 2008).

## O.1  Communities and Crime Data Set

This data set contains 1994 samples with 122 predictors which might have plausible connection to crime, and the attribute to be predicted (Per Capita Violent Crimes). In the preprocessing step, any predictors with missing values are removed and all the predictors and the attribute to be predicted are standardized to have zero mean and unit standard deviation. The preprocessed dataset contains $d = 100$ predictors and $n = 1994$ samples.

The optimization problem (6) is solved for $\lambda_n = 0.15$ and $\gamma$ is chosen to be $\frac{\max(\mathbf{y}) - \min(\mathbf{y})}{2}$. As the problem is invex, any algorithm which converges to a stationary point can be used to solve the problem. We used an alternate optimization algorithm (See Appendix K) which converges to an optimal solution.

**Main results.** Based on the support (non-zero entries) in the recovered $w$, we found that the following are the most important predictors of Per Capita Violent Crimes:

1. PctHousNoPhone: percentage of occupied housing units without phone
2. PctNotHSGrad: percentage of people 25 and over that are not high school graduates
3. PctLess9thGrade: percentage of people 25 and over with less than a 9th grade education
4. RentLowQ: rental housing - lower quartile rent

We also recovered the hidden sensitive attribute with 816 instances of positive bias ($z = +1$) with mean crime rate $0.8002$ and 1178 instances of negative bias ($z = -1$) with mean crime rate $-0.5543$. By plotting data with two of the most important predictors (PctHousNoPhone, PctNotHSGrad), we clearly see the existence of two groups (Figure 5). Our Mean Squared Error (MSE) is 0.0265. Chzhen et al. (2020) can be checked for comparison with other state-of-the-art methods (12 methods of 3 different types) where only the Kernel Regularized Least Square method (MSE=$0.024 \pm 0.003$) and the Random Forests method (MSE=$0.020 \pm 0.002$) perform better than our method in terms of MSE but suffer heavily in terms of fairness. Other methods incur MSE in the range between $0.028 \pm 0.003$ to $0.041 \pm 0.004$.

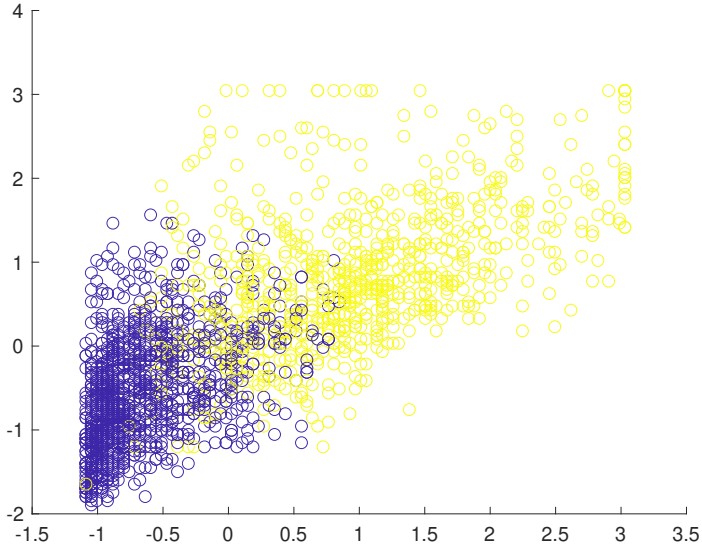

Figure 5: Clusters in Communities and Crime Dataset

## O.2  Student Performance Data Set

This data set contains 649 samples with 33 demographic, social and school predictors and the attribute to be predicted (grade in the Portuguese Language course). The data set contains some

categorical variables which are converted to numerical variables using dummy encoding (thus increasing the number of predictors). Two columns containing partial grades were removed from the data set. In the preprocessing step, all the predictors and the attribute to be predicted are standardized to have zero mean and unit standard deviation. The preprocessed dataset contains $d = 39$ predictors and $n = 649$ samples.

Similar to subsection O.1, the optimization problem (6) is solved for $\lambda_n = 0.15$ and $\gamma = \frac{\max(\mathbf{y}) - \min(\mathbf{y})}{2}$.

**Main results.** The following are the most important predictors of grades in the Portuguese Language course:

1. school: student's school
2. failures: number of past class failures
3. higher: wants to take higher education

We also recovered the hidden sensitive attribute with $420$ instances of positive bias ($z = +1$) with mean grade $0.2305$ and $229$ instances of negative bias ($z = -1$) with mean grade $-0.4227$. Our Mean Squared Error (MSE) is $0.0494$. Chzhen et al. (2020) can be checked for comparison with other state-of-the-art methods (12 methods of 3 different types) where none of the methods performs better than our method in terms of MSE (range between $3.59 \pm 0.39$ to $5.62 \pm 0.52$).

## O.3 Discussion.

While our analysis identifies two groups with bias in both data sets, it cannot only be attributed to the most important recovered predictors. Recall the "red-lining" effect (Calders, 2010) where there might be other correlated predictors which can facilitate indirect discrimination. For example: in the Communities and Crime data set, annual income could be correlated with PctHousNoPhone and similarly in the Student Performance data set, parents' educational qualification could be correlated with student's willingness to go for higher education. Our analysis does not ignore such factors. In fact, even after taking the red-lining effect into the consideration, our method is able to identify two groups with bias.