# OpenReview forum: "Fair Sparse Regression with Clustering: An Invex Relaxation for a Combinatorial Problem"
_NeurIPS.cc/2021/Conference — NeurIPS 2021 Spotlight_

### Official Review · Reviewer_iiXG · 2021-07-14

**Rating:** 6
**Confidence:** 3

**Summary:**

The paper considers sparse regression with a hidden binary attribute modeling unknown bias (as in lack of fairness).  This is in contrast to most other work on fairness where the protected attribute is known, and used to impose fairness constraints.  The paper presents an invex relaxation (bi-convex generalization of convexity without local minima), proves its correctness, and makes claims that it can exactly identify hidden bias, and does so at no cost to prediction accuracy.

**Ethical Concerns:**

Please see the main review.  From my understanding the proposed method will always detect hidden bias (whether or not there is one in the data) -- and there are no suggestions for statistical tests to check if it's real.

**Ethics Review Area:**

["Discrimination / Bias / Fairness Concerns", "Inadequate Data and Algorithm Evaluation", "Inappropriate Potential Applications & Impact  (e.g., human rights concerns)"]

**Limitations And Societal Impact:**

No -- from what I understand the paper goes into dangerous ethical territory.  Please see main review.  I would advise to think about less mathematically savvy practitioners who may use your method, and devise statistical tests to check if the discovered bias is real or not.

**Main Review:**

The paper is very carefully done from the theoretical point of view (formulation and analysis), with valuable contributions to invex relaxations of combinatorial problems.  I believe such relaxations could be used in other applications as well, e.g. perhaps  for 'mixed liner regression' or related problems.  I really enjoyed both the formulation, and the rich toolkit involved the analysis.

However, the application to fairness seems quite problematic, and this is the first paper where I would worry that it may indeed lead to an adverse societal impact.  Experiments and practical implications seem to be an after-thought of the new methodology.
I am completely on board with the need to ensure fairness in ML methods, but this paper shows that there is also a real danger of over-reach if such approaches are used without appropriate care or considering practical implications. (Please correct me if you feel I misunderstood the setting).

Consider a generic (sparse) linear regression problem y = X w + noise , where the data was simulated and includes no bias (no gamma*z term).  About half of the training data-points would lie above the predicted y_hat,  and about half would lie below the y_hat.   I believe that if you apply the proposed invex formulation,  it will still find some clustering which will roughly align with under and over predicted points,  and after 'de-biasing' shrink the clouds of points to be closer to linear model model predictions.  My intuition is that if you take a perfectly homogeneous Gaussian cloud of points, and apply a clustering method (say spectral clustering),  these points will still be split into two clusters -- perhaps two semi-circles.  So whether or not there are real clusters -- the method will find them, or in other words, the method will find bias even if there's none.  I'm being overly dramatic here -- but it's basically a first step on the pathway to communism -- where everyone is equal and perfectly predictable by the model.  I could easily imagine how a team from a consulting company (say Accenture or IBM) would use the proposed approach and advise the government of say Kazakhstan or Papua-New-Guinea to detect hidden biases.  Another related issue -- in cases with real bias in the data:  some biases are bad (e.g subjectively prioritizing majority groups over minorities) -- but some biases are good -- say kids who went to a math summer camp vs. kids who watched TV.   So if the latent labels are hidden -- then there may be no way to separate good bias from bad bias.  This also goes into murky ethical waters.

To be clear, if the data was generated according to the model in (1) with large gamma*z terms,  i.e. where gamma is large enough compared to the standard deviation of the noise term,  then the method makes sense, and indeed it should recover both the z_i's and the sparse w.  So what's missing is some statistical significance that shows that the clusters are "real".  Also strong claims throughout the paper that the proposed method 'exactly recovers the hidden attribute of each sample' need to be qualified.  There are n observations, and s+n unknowns (non-zero w's and all z's), so in case  gamma is comparable to noise-sigma,  clearly the problem is not exactly identifiable.  This basic intuition is not shared in the paper -- instead it makes assumptions on support, and mutual incoherence, and not relating them to what may make sense (or can in fact be tested) in practice.

Other comments:
1) The inclusion of debiasing / fairness constraint has no adverse impact on the performance, despite what the rest of the literature concluded.  (last sentence on page 2). As I understand there's no paradox here -- you're just not using the same notion of performance as other papers on fairness.  The usual baseline is || y - Xw||  on biased data (say on training set). Clearly by enforcing constraints this performance (MSE) should get worse.   In your case from what I understand you're using || y - Xw - gamma *z|| as "performance" -- which is akin to how well your debiasing would compare to an oracle "debias", but not the original MSE with biased data.
2) Real data experiments are lacking (none in the main body), and even in the appendix it's unclear exactly what was done and what have you uncovered with z_i's.

Minor comments:
1) page 2 "While ... Lasso ... provides an estimate of the regression parameter vector, they do not fit the model accurately,  as they fail to consider any fairness criteria"...   I think there's a confusion of accuracy and fairness -- the method may be accurate on biased data,  but unfair,  by making it fair -- it'll be less accurate on the original (biased) data.
2) "our method recovers the exact hidden attributes for each sample and thus provides an exact measure of bias between two different groups".   I'd suggest adding some qualifiers to such strong statements.
3)  You should explicitly mention whether X and z are allowed to be correlated in the model description on page 3.
4) We will assume that as the number of samples increases the noise in the model gently decreases.   Why does this make sense (beyond a mathematical convenience for proofs)?
5) You make support assumptions on w -- but in reality w's are never exactly sparse -- more likely near-sparse, say like power law.  Do your proofs still hold?


**Needs Ethics Review:**

Yes

**Time Spent Reviewing:**

4

---

> ### Author Response · Authors · 2021-08-11
> **Response to Reviewer iiXG**
>
> It seems like the reviewer has let their personal beliefs come in the way of the objective assessment of our work. In contrast, all other reviewers (muDS, jdvv, SRWv) have recognized and acknowledged our contributions and impact in their reviews. As noticed by all the other reviewers, our use of invex relaxation method for solving the inverse problem is novel. Our method can be generalized to solve other NP-hard combinatorial problems which include but are not limited to learning Bayesian networks, inference in structured prediction, and community detection. Reviewer jdvv and SRWv have also praised our presentation and soundness of our technique. We would also like to mention reviews from Reviewer SRWv has succinctly summarized our contributions in four categories: originality, quality, clarity and significance. Unlike prior work in fairness [Calders et al., 2013; Agarwal et al., 2019; Fitzsimons et al.,2019, Berk et al.,2017, Chzhen et al.,2020], our proposed method is original as it does not have access to sensitive attribute in the data. For this new setting, we have formulated a novel optimization problem and provided an invex relaxation for the same. Reviewer SRWv also notices that claims of our paper are well supported mathematically and empirically by computational experiments on the real-world datasets. Reviewer SRWv has also recognized the importance of fairness in a wide range of applications where it is important to avoid biased decisions and our contributions to the field for the scenario when there is a hidden biased attribute.
>
> Notice that none of the other reviewers have raised any ethical concerns. We strongly believe that detecting bias (without classifying it as good or bad) is a scientific task which should be left to the scientists. Making sense of the detected bias is the job of the field expert. While we all can have personal opinions on this topic, it is the job of the experts to take the final call.
>
> Now, we will address questions raised by the reviewer.
>
> - The second paragraph ("However...setting).") gives no well articulated specific critic to the paper. Rebuttal phase is for the discussion on academic issues pertaining to the paper. Reviewer alludes to some vague preconceived personal opinions in this paragraph but does not provide any specifics which we can respond to.
>
>
> - "Consider a generic (sparse) linear regression problem y = X w + noise , where the data was simulated and includes no bias (no gamma*z term)... shrink the clouds of points to be closer to linear model model predictions."
>
> Our method requires gamma to be greater than 0 (Line 96). On a pure technical point of view, it is not fair to ask what happens when technical conditions required for our analysis do not hold.  The analysis will, of course, not apply -- just like any other technical analysis. Note that our algorithm will still run but our guarantees may not hold.
>
> - "My intuition is that if you take a perfectly homogeneous Gaussian cloud of points, and apply a clustering method (say spectral clustering), these points will still be split into two clusters -- perhaps two semi-circles. So whether or not there are real clusters -- the method will find them, or in other words, the method will find bias even if there's none."
>
> Keeping our technical requirement aside (as described in our previous response), any clustering algorithm will always recover clusters whether they exist or not. Given this reasoning any paper on fairness [Calders et al., 2013; Agarwal et al., 2019; Fitzsimons et al.,2019, Berk et al.,2017, Chzhen et al.,2020] and clustering should get rejected.
>
> - "I'm being overly dramatic here -- but it's basically a first step on the pathway to communism -- where everyone is equal and perfectly predictable
> b y the model. I could easily imagine how a team from a consulting company (say Accenture or IBM) would use the proposed approach and advise the government of say Kazakhstan or Papua-New-Guinea to detect hidden
> biases."
>
> This statement shows some specific personal political views, and also goes against the inclusive nature of an international research community. It is unfortunate that we have to respond to such comments in a conference like NeurIPS.
>
> - "Another related issue -- in cases with real bias in the data: some biases are bad (e.g subjectively prioritizing majority groups over minorities) -- but some biases are good -- say kids who went to a math summer
> camp vs. kids who watched TV. So if the latent labels are hidden -- then there may be no way to separate good bias from bad bias. This also goes into murky ethical waters."
>
> We address a broader question. Consider an oracle which detects bias in data. It shows the result that group A has positive bias and group B has negative bias. Can we say anything about bias being good or bad? The answer is no. What if we say Female employees receive less salaries than  Male employees with same qualifications. Can we now say if this bias is bad? Yes, we can because now we have more context and we are thinking like a "field expert". The method can only detect the bias, it is the job of the field expert to make sense of it. We understand the concern over bias being good or bad but we will never get to that question unless we have a method which detects bias. Refusing to even do that is nothing but burying our head in the sand.
>
> - On the cases when gamma is small
>
> The magnitude of gamma impacts our technical results and as clearly mentioned in Theorem 1, the sample complexity depends upon gamma. In particular, for small gamma we need large number of samples. Given that the technical conditions satisfy, our method recovers the labels exactly with high probability. While it is interesting to think about doing a statistical analysis when technical conditions are not satisfied, it is beyond the scope of this work.
>
>
> - On defining performance criteria:
>
> Our true performance criteria is recovery of support of w and labels (z). The reviewer is talking about loss. Let us put it in another way. Which method explains the response variable more accurately in biased data? Note that after detecting z, it only makes sense to use ||y - Xw - gamma * z|| as loss. It seems unfair to blame us for using a refined loss function. If anything, using ||y - Xw|| as performance criteria should be treated as the limitation of other methods.
>
>
> - On real world experiments:
>
> We have two experiments on real world dataset which are mentioned in main paper with details deferred to Appendix O (the details span over two pages). We describe the experiments in detail and accompany them by providing bulleted main results and a discussion at the end. We do not understand how reviewer came to the conclusion that real world experiments are lacking.
>
>
> - On accuracy in biased data:
>
> Assessment of the reviewer is incorrect. If the ground truth follows generative model of equation (2), Lasso will not be accurate on biased data and our method will be.
>
> - On adding qualifiers to claims:
>
> In every technical part of the paper, all our statements and claims are well qualified (All the Assumptions, Lemmas, Theorems and Corollaries).
>
>
> - On correlation between X and z:
>
> While X is a random variable, z is a deterministic quantity.
> Therefore, it does not make sense to talk about correlation between X and z from a statistical point of view. Correlation is between two random variables and not between a constant and a random variable.
>
>
> - On decreasing noise variance:
>
> It seems strange to criticize a decay of rate O(1/sqrt(log n)). The decay is so gentle that it can be treated as a constant. It is standard practice to treat such gentle rates as constants in theoretical CS. For example: \tilde(O) ignores all the logarithmic factors as they grow very slowly.
>
>
> - On using sparse w
>
> It is impossible to exactly model w in reality. Even the power law mentioned by the reviewer is only an approximation. There is a large amount of work which assume sparse w (subset selection in regression [Miller et al., 1990] compressed sensing [Donoho, 2006; Candes et al., 2004], structure estimation in graphical models [Meinshausen et al., 2006], sparse approximation [Devore et al., 1993], signal denoising [Chen et al., 1998] etc.). Analysis near-sparse w setting seems interesting but it is beyond the scope of this work.

---

> > ### Comment · Reviewer_iiXG · 2021-08-11
> > **Reply to authors response**
> >
> > Thank you for the detailed response. As I stated in my original review, the paper is strong technically and proposes valuable
> > methodological contributions. If NeurIPS was purely a mathematical conference, I'd give the paper a high rating. I also fully
> > agree that the techniques developed here should have other interesting applications beyond fairness.
> >
> > NeurIPS, however, is no longer a purely mathematical conference, and the tools proposed and developed in published papers
> > receive significant attention from businesses and government organizations and are often used to make consequential decisions.
> > I believe this is the reason that 'ethics' and 'societal implication' evaluation has been introduced in the first place. Hence, it is the duty
> > of the authors to alert potential users to various issues that may arise when using the technique in practice.  The ethics
> > reviewer agrees that there's a real potential for misuse of the proposed technique, and the authors should clearly discuss
> > practical caveats and limitations of applying your work in practice.  I agree with that assessment -- I will gladly increase
> > my score (to reflect the technical merits of the paper) if the paper clearly states the limitations and caveats of using the
> > approach in practice.
> >
> > Specific comments:
> > 1) "Our method requires gamma to be greater than 0 (Line 96). On a pure technical point of view, it is not fair to ask what
> > happens when technical conditions required for our analysis do not hold."  ...  "The magnitude of gamma impacts our technical
> > results and as clearly mentioned in Theorem 1, the sample complexity depends upon gamma".
> >
> > This is a clear example of ignoring any practical implications. When a data-scientist gets a novel data-set -- how does he/she
> > verify if gamma is greater than 0?  How do they know how many samples are sufficient to believe the results? How do they
> > get any confidence that the proposed method found a real bias, or whether it's a statistical artifact?
> >
> > I would expect a very clear discussion of these issues, as you are boldly positioning your paper as a practical tool, with statements
> > like "Our method can detect bias in the system. In particular, our method recovers the exact hidden attributes for each sample
> > and thus provides an exact measure of bias between two different groups.".
> > Such purely mathematical statements that hold under very specific assumptions, and are impossible to observe / verify in real data -- can be quite dangerous when applied in the wild.  Note, I'm not asking you to solve statistical significance issues in the revised paper, it is
> > indeed a hard problem, and a topic of ongoing research,  but I'm asking to clearly acknowledge that in lieu of any notion of statistical significance -- one has to be really cautious of trusting and interpreting the results of your method.  In particular, please add some form of the discussion in your response that trying to regress the discovered latent variables against some existing traits (like male vs. female) as provided by field expert -- would be essential if such analysis were to be used on real data.
> >
> > 2) Our true performance criteria is recovery of support of w and labels (z). The reviewer is talking about loss.
> >
> > You define one performance metric,  other papers also use other metrics.  The conclusion that you reach in the paper is specific
> > to your metric.  I'm asking to add a caveat -- that the conclusion would be different if you use another metric (which to
> > my knowledge is more commonly used in the fairness literature).
> >
> > 3) I do apologize for naming specific countries -- I do not have any specific political views, but unfortunately some organizations
> > (such as governmental agencies, especially in countries with more limited resources) that would like to employ machine
> > learning-based models may not have the resources or the expertise to fully understand the assumptions, limitations and
> > implications of the proposed techniques. It is thus the duty of the authors to inform them about these.

---

> > > ### Author Response · Authors · 2021-08-18
> > > **Response to Reviewer iiXG**
> > >
> > > We welcome NeurIPS's decision to include ethics and societal implication evaluation to the review process. We propose to add the following discussion to the "Societal Impact Section" in the final version of the paper:
> > >
> > > "Fairness in machine learning is an active field of research. As it has the potential to affect the basic well being of our society, it should always be handled in consultation with field experts. This is especially important when one moves away from the theoretical setting and tries to apply fairness algorithms to real world data where the validity of technical assumptions cannot be easily verified. As with any algorithm in prior literature ([Calders et al., 2013; Agarwal et al., 2019; Fitzsimons et al.,2019, Berk et al.,2017, Chzhen et al.,2020]), one has to be extra cautious when interpreting the results of our method as well. In particular, we emphasize that our method does not characterize the nature of the recovered bias - good or bad. The nature of the bias should be determined by the field expert. We highly recommended to seek the field expert's input for interpreting the results in a real world setting. We advise to regress the discovered hidden attribute against some existing predictors provided by a field expert."
> > >
> > > We are open to discuss more suggestions from reviewers to add to this section.
> > >
> > > Below we answer specific questions from the reviewer.
> > >
> > > - We have addressed the first concern in our draft for the "Societal Impact Section".
> > > - We will acknowledge in the final version that different settings may employ different loss functions. We use mean square error with $\gamma z$ because it makes sense in the setting of our problem (Compare eq 3 and 4).  This formulation allows us to recover the correct support as well as the correct labels.

---

> > > > ### Comment · Reviewer_iiXG · 2021-08-22
> > > > **reply to authors**
> > > >
> > > > Thank you, I increased the rating. I do hope the paper will follow through clearly acknowledging the limitations and caveats for applications in practical settings.  The strength of the paper is in the mathematical formulation, the application is very stylized.  Also clarifying that the surprising conclusion 'there is no cost for de-biasing' -- specifically depends on your chosen error metric -- will avoid a lot of potential confusion.

---

### Official Review · Reviewer_SRWv · 2021-07-15

**Rating:** 7
**Confidence:** 3

**Summary:**

The paper considers the task of sparse regression on a biased dataset where the bias depends on a hidden binary attribute. A new optimization problem is formulated which is then relaxed into an invex optimization problem. The authors then prove that their invex fair lasso formulation correctly recovers the hidden attributes and finds a regression parameter vector with the same support as the true parameter vector. This is done by constructing a pair of primal and dual variables that have the desired properties and fulfill the KKT conditions. In the experimental section, the theoretical results are validated on synthetic data. Moreover, the method is evaluated on two real world datasets where the hidden attributes are recovered in both cases.

**Limitations And Societal Impact:**

The authors discuss the assumptions made and therefore the limitations of the given approach in Section 4. There is no negative societal impact of the proposed work.


**Main Review:**

Originality:

The proposed method differs significantly from previous approaches as it assumes no direct access to the sensitive attribute in the data. Instead, the bias depends on a hidden attribute. For this new setting, it proposes a novel optimization problem which is then relaxed into an invex optimization problem. Previous work is discussed and cited adequately.

Quality:

The claims of the paper are well supported. First, the paper shows theoretically that the pair of primal-dual witnesses correctly recovers the hidden attributes and with high probability yields a parameter vector with the correct support. Those theoretical claims are then again validated experimentally on synthetic data. Finally, on two real-world data sets it is shown that the method also works in practice and can be used to recover groups with bias.

Clarity:

The paper is clearly written and generally well-organized. By nature, the content of Section 4 is more technical and therefore a bit harder to read. However, the authors do a good job in organizing it in different subsections so that it can be followed easily. On the downside, while the paper contains the presentation of the approach and the proof of the optimality of the obtained solution, the description of how to actually solve the optimization problem is delegated to the supplementary material (Appendix K). The paper could be more self-contained by delegating parts of the proofs in Section 4 to the appendix (while keeping the general structure and outline of Section 4) and filling up the gained space by an abridged version of Appendix K.

Significance:

The problem of fairness is an important topic since there are a wide range of applications where it is important to avoid biased decisions. The paper makes an important contribution to the field by considering the scenario where there is a hidden biased attribute. Additionally, the topic of invexity has recently gained some attention in the community. For these reasons, the paper will likely be of interest to the community.

---
After rebuttal:

I have read through the responses by the authors as well as the feedback of the other reviewers. I now believe that the ethical concern raised by reviewer iiXG as well as the ethics reviewers is justified. However, I believe that these concerns can be sufficiently addressed if the authors add a section discussing the societal and ethical implications in the paper, incorporating the changes proposed by the ethics reviewers. Therefore I will keep my score.


**Time Spent Reviewing:**

4

---

> ### Author Response · Authors · 2021-08-11
> **Response to Reviewer SRWv**
>
> We would like to thank the reviewer for their feedback and a thorough review. We will definitely consider your suggestion to replace some portion of Section 4 with a discussion on Appendix K.

---

### Official Review · Reviewer_jdvv · 2021-07-17

**Rating:** 6
**Confidence:** 3

**Summary:**

A problem of fair sparse linear regression is addressed in this paper, where bias depends upon a hidden binary random variable. This hidden r.v. can be thought of as a clustering assignment combining sparse regression. An optimization problem is formulated by the authors as a mixed-integer linear programming originally, then, a continuous relaxation is used to reformulate it into an invex optimization problem. The theoretical results mainly focus on the solution's uniqueness of the invex problem by verifying Karush-Kuhn-Tucker conditions and primal-dual certificates. In the experiments, the proposed method shows the ability to correctly recover the support of the regression parameter vector, and the correct clusters.


**Limitations And Societal Impact:**

The author's work might have a potential positive societal impact, which is aimed to identify the bias in decision-making in human society.

**Main Review:**

The topic of this paper is of significant interest to the machine learning community. This framework could potentially shed a light upon a novel strategy for several NP-hard combinatorial problems including learning Bayesian networks, inference in structured prediction, and community detection. The technique is presented well and technically sound. My concerns are listed below -

1. The proposed algorithm aims to solve the regression problem while identifying the bias for each subgroup. I am wondering how the authors attain prior knowledge of the number of underlying subgroups? The paper discussed two subgroups in which case the invexity can be utilized in the optimization as a learning guarantee, but I am wondering if more discussion can be expanded on the real cases when there are more than 2 subgroups with different biases.

2. The proposed algorithm assumes that the linear/nonlinear sparse model is universally adopted among both subgroups, which sounds not realistic. It would be more appreciated to think about certain error tolerance of the model discrepancy among subgroups, while preserving the good property of the model such as invexity. E.g., if the generative model parameters of the two subgroups are w_1 and w_2, respectively, how sensitive the model is regarding the clustering results when w_1 differs with w_2?

3. By using a DAG model, one can formulate the problem with one additional plate as a Bernulli r.v. in the probabilistic graphical model if the solution's uniqueness is not the central concern. Some empirical comparisons such as DAG models can be added to the experiments.

4. I am not sure why the paper has no conclusion, but I would suggest the authors consider do conclude the work, which can give more important insights and discussions on the future direction.

**Time Spent Reviewing:**

24

---

> ### Author Response · Authors · 2021-08-11
> **Response to Reviewer jdvv**
>
> We thank the reviewer for their feedback and questions. We have addressed them here. We hope that this discussion clarifies all the questions they have and leads to further improvement of our paper.
>
> - On attaining prior knowledge of number of subgroups and extending results to multiple subgroups:
>
> The number of sub-groups can be determined by the help of a field-expert. One possible way to generalize our model to work with m subgroups is by using one-hot encoding of length m for zstar for each sample, i.e., if the sample belongs to i-th subgroup, the i-th entry of zstar will be 1 and all other entries will be 0 and an m-dimensional vector for gamma.
>
> In the general case, z will be nxm matrix and gamma will be mx1 vector. Analogous to our paper, a general SDP type relaxation for multiple subgroups can be done using SDP-1 type formulation from [Amini et al., 2016].
>
> [Amini et al., 2016] : Amini, A. A., & Levina, E. (2018). On semidefinite relaxations for the block model. The Annals of Statistics, 46(1), 149-179.
>
>
> - On using same linear sparse model for both subgroups:
>
> It is hard to talk about fairness when w_1 is not same as w_2. All the papers [Calders et al., 2013; Agarwal et al., 2019; Fitzsimons et al.,2019, Berk et al.,2017, Chzhen et al.,2020] in Table 1 use different w. Our proposed model works when two candidates with same qualification but from two different subgroups are discriminated against each other. There are many real life decision problems where this makes sense. For example: detecting salary difference in male and female employees with same qualification, detecting difference in incarceration rate in a black and a white neighborhood with similar crime rate. In these kind of examples, w will be the same for both subgroups. The proposed extension by reviewer is an interesting idea and we can think more about it in our future work.
>
> - On comparing with a DAG model:
>
> Taking z to be a Bernoulli random variable, we studied reviewer's suggestion, using an EM method. After careful derivations, our conclusion is that such a method will be equivalent to Standard LASSO in eq.(3) and thus, will not provide any advantage over our method.
>
> - On paper having no conclusion:
>
> We will add a conclusion in the final version of the paper. It will look as below.
>
> "
> In this paper, we provide a novel formulation of an invex fair LASSO which incorporates fairness constraints into the standard LASSO problem without compromising on the performance. We show that invexity of our optimization problem allows for a tractable solution. We provide provable theoretical guarantees for our solution and further validate them by computational experiments. The sample complexity of our method is polynomial in terms of sparsity and logarithmic in terms of the dimension of true parameter. Our method helps to identify and subsequently remove bias in the sparse regression model. In future, it will be interesting to study invex relaxation of other models of fairness. Since set of invex functions subsumes convex functions, it will enable us to tackle a larger set of problems.
> "

---

### Official Review · Reviewer_muDS · 2021-07-21

**Rating:** 7
**Confidence:** 3

**Summary:**

This paper studies the inverse problem of fair sparse regression. The corresponding optimization problem contains a continuous variable and discrete variable, and it is combinatorial. The authors consider a novel invex relaxation of it. They prove that there exists a unique solution to the invex problem，and the true variables of the model can be recovered exactly.

**Ethics Review Area:**

["I don’t know"]

**Limitations And Societal Impact:**

Yes.

**Main Review:**

This paper is generally well written. The use of the invex relaxation method for solving the inverse problem is novel. The authors also prove that the sample complexity of the invex relaxation method is logarithmic in terms of the dimension of the regression parameter vector. They also provide sufficient details of their methods, including the technical proofs, the alternating minimization method, and the experimental details. I believe that the proposed method could be applied to other combinatorial problems and continue to be useful.

However, I have a few concerns.
W1. The proposed invex relaxation method could be of theoretical interest only since it requires solving an SDP problem which could be a daunting task since its computational complexity is very high.
W2. The number of samples and dimensions of the data set used in the experiments are too small. For example, they only consider n<500 and d<1000 in the experiments.
W3. The CPU time spent for solving the invex problem is not reported.
W4. The authors do not compare their method with other convex or nonconvex relaxation methods in their experiments.




**Time Spent Reviewing:**

4

---

> ### Author Response · Authors · 2021-08-11
> **Response to Reviewer muDS**
>
> We thank the reviewer for their feedback and questions. We have addressed them here. We hope that this discussion clarifies all the questions they have and leads to further improvement of our paper.
>
>
> - On proposed invex relaxation only being of theoretical interest:
>
> With recent advancements in SDP solvers, it is now possible to solve high dimensional SDPs quiet efficiently. [Yurtsever et al., 2021] propose algorithms which can tackle SDPs with more than 10^14 entries. Therefore, solving SDP is not an issue for our problem.
>
> Besides, while invex relaxation and subsequent proofs for exact label and support recovery do not readily provide an efficient algorithm to solve the problem, they do ensure that any algorithm that finds a stationary point will find the optimal solution. We believe that this in itself is a good first step towards solving a seemingly intractable problem.
>
> [Yurtsever et al., 2021] : Yurtsever, A., Tropp, J. A., Fercoq, O., Udell, M., & Cevher, V. (2021). Scalable semidefinite programming. SIAM Journal on Mathematics of Data Science, 3(1), 171-200.
>
> - On computational experiments with bigger n (> 500) and d(> 1000) and providing CPU time:
>
> As mentioned previously, [Yurtsever et al., 2021] allows us to tackle SDPs with more than 10^14 entries.
>
> As a proof of concept, we have solved our problem for n = 5000, d = 10000 and s=20 entries. Our method runs in 31.35 seconds in a Linux machine with 8GB RAM and recovers the labels exactly.
>
>
> - On comparison with other methods:
>
> Our experiments on synthetic data verify our theoretical results. For real world experiment, we do compare our results with other state-of-the-art methods (Appendix O, Line 770, 792). The details of these methods can be found in Chzhen et al. (2020).

---

### Review · Ethics_Reviewer_faDi · 2021-08-10

**Recommendation:**

A clearer discussion of limitations focusing on (a) how finding bias in every dataset might not be a good idea, (b) the real-world limitations of assigning positive bias and negative bias to every point, would be immensely valuable.

**Ethical Issues:**

Yes

**Ethics Review:**

This is a very well-written and well-executed paper that technically achieves quite a lot (I particularly appreciate the authors trying to demonstrate this on real world datasets) – but I am not sure the proposed fair regression problem is a realistic one. My understanding is that the authors try to simultaneously solve sparse linear regression and a clustering problem – the clustering in part in particular focuses on assigning data points a positive bias or a negative bias value. The core technical achievement (in my reading) seems to be doing multiple optimizations in parallel, with theoretical guarantees. However, applying this technical apparatus to a fairness task seems rather contrived, like a very good hammer looking for a nail.

My core worry is particularly in the rigid $z = +1$ or $z = -1$ assignments. For instance, in Appendix O, when these assignments are done to each data point in the Communities and Crime Dataset, what does it qualitatively mean? What kinds of data points are being considered positive bias vs. negative bias? If something like this was to published in a software package, and someone could use only the $z$ assignments to ostensibly implement fairness (and not the complete sparse regression task), could it be harmful? I wish the authors could provide some descriptive data analyses on what features do these $z$ most highly correlate with in their real world experiments, and what are the kinds of cases where such a bias assignment is useless on its own. I think Reviewer iiXG's worry about finding bias where there is none also stems from this system of rigid $z$ assignment. While that is a real concern, what I'm highlighting is a possible misuse of the authors' technical contributions. A bit of descriptive data analyses and carefully peppered caveats and limitations throughout the paper would ensure that a misinformed reader doesn't end up using this methodology for the wrong reasons.

The paper and the reader could benefit from a higher level discussion of the limitations of this approach.

---

> ### Author Response · Authors · 2021-08-18
> **Response to Ethics Reviewer faDi**
>
> We welcome NeurIPS's decision to include ethics and societal implication evaluation to the review process. We propose to add the following discussion to the "Societal Impact Section" in the final version of the paper:
>
> "Fairness in machine learning is an active field of research. As it has the potential to affect the basic well being of our society, it should always be handled in consultation with field experts. This is especially important when one moves away from the theoretical setting and tries to apply fairness algorithms to real world data where the validity of technical assumptions cannot be easily verified. As with any algorithm in prior literature ([Calders et al., 2013; Agarwal et al., 2019; Fitzsimons et al.,2019, Berk et al.,2017, Chzhen et al.,2020]), one has to be extra cautious when interpreting the results of our method as well. In particular, we emphasize that our method does not characterize the nature of the recovered bias - good or bad. The nature of the bias should be determined by the field expert. We highly recommended to seek the field expert's input for interpreting the results in a real world setting. We advise to regress the discovered hidden attribute against some existing predictors provided by a field expert."
>
> We are open to discuss more suggestions from reviewers to add to this section.
>
> Below we answer specific questions from the reviewer.
>
> - The reviewer makes a good point about the binary assignments of z. As we mention in Line 49-52, assigning z to +1 or -1 is not equivalent to identifying bias due to one (and only one) sensitive attribute. The role of field expert becomes extremely important here to get more insight into the findings. The correlation between important attributes and the assignments z may provide some (but perhaps not all) insights but it is best left to the field expert.
>
> We conducted real world experiments in Appendix O (since our main contribution is theoretical). In those datasets, we found the following correlations:
>
> Communities and Crime Data Set
> Response: Per Capita Violent Crimes
> PctHousNoPhone: percentage of occupied housing units without phone, Correlation: 0.7699
> PctNotHSGrad: percentage of people 25 and over that are not high school graduates, Correlation: 0.6589
> PctLess9thGrade: percentage of people 25 and over with less than a 9th grade education, Correlation: 0.6036
> RentLowQ: rental housing - lower quartile rent, Correlation: -0.5123
>
> Student Performance Data Set
> Response: Grade in the Portuguese Language course
> school: student’s school, Correlation: 0.9899
> failures: number of past class failures, Correlation: -0.1261
> higher: wants to take higher education, Correlation: -0.1428
>
> Note that high (or low) correlation may not mean causation. It is important to seek the help of the field expert for interpretation. We refer the reviewers to Appendix O.3 for a discussion on this.

---

> > ### Comment · Ethics_Reviewer_faDi · 2021-09-03
> > **Response**
> >
> > I'm uncomfortable with using "field expert" as a lazy evasion of responsibility and I would suggest the authors don't put dismissive language like that in the paper. The authors cannot provide a toolkit to do a fair regression task, and then shrug at the end and put the responsibility of interpreting _their own_ intermediate bias assignments through some abstract field expert. I'm glad ethical reviews are part of the pipeline, and I hope the authors can constructively take my criticism.
> >
> > The core possible harm that the paper might contribute to the ML community is this (line 39-41): "All the above work assume access to the sensitive attribute in the training samples and provide a framework which are inherently fair. Our work fundamentally differs from these work as we do not assume access to the sensitive attribute". What the authors do instead is to assume a sensitive attribute, which has no real world analogue. While this might in some cases cause the regression to be "fairer" (again based on an abstract notion based on z = +1/-1), it is by no means something that is actually in the dataset, or interpretable by a practitioner (engineer/data scientist) of this fair regression. It is the responsibility of the authors as rational scientists to add these caveats to their text, so a lazy practitioner cannot abuse these methods.
> >
> > Working off of what the authors have written, I would suggest that they add the following to the paper -- with a bold limitations heading (there is currently none in the paper):
> >
> > "Fairness in machine learning is an active field of research. As it has the potential to affect the basic well being of our society, it should always be used with caution. This is especially important when one moves away from the theoretical setting and tries to apply fairness algorithms to real world data where the validity of technical assumptions cannot be easily verified. As with any algorithm in prior literature ([Calders et al., 2013; Agarwal et al., 2019; Fitzsimons et al.,2019, Berk et al.,2017, Chzhen et al.,2020]), one has to be extra cautious when interpreting the results of our method as well. In particular, we emphasize that our method does not characterize the nature of the recovered bias - good or bad. Our proposed hidden attribute could be seen as a proxy for algorithmic bias, but one that exists outside the features of the data itself. To interpret the meaning of such an attribute, we advise consulting a domain expert, and to regress the discovered hidden attribute against some existing predictors provided by the expert. We also caution practitioners that such a notion of hidden bias would naturally not extend to every task, and should not be used as a silver bullet to justify fairness in socially significant contexts."
> >
> > I do believe that the core technical contribution of this paper is technical, which could be valuable at a venue like NeurIPS. I also believe that the application of the proposed optimization to a fairness context is rather contrived. But often in science, it is difficult to find the right application for an interesting technique -- and I hope the authors here understand that adding caveats here is their scientific responsibility.

---

> > > ### Author Response · Authors · 2021-09-04
> > > **Response to Ethics Reviewer faDi**
> > >
> > > We appreciate the suggestion for changing our initial societal impact statement, including the bold limitations heading. We will definitely take this into account.

---

### Review · Ethics_Reviewer_Ag79 · 2021-08-11

**Recommendation:**

I believe it is possible to address these issues in a revision, but I would not be comfortable taking it on faith, given the authors' dismissive response to reviewer iiXG that the authors will address these issues.   I would therefore recommend that the authors submit a revision, or that the paper be shepherded to ensure that ethical concerns are addressed before the paper is published.

**Ethical Issues:**

Yes

**Ethics Review:**

I agree with the assessment of reviewer iiXG, and have some further comments:

1) It is important that the authors make explicit under what definition of bias their framework operates, in what context that type of bias would lead to harms, and what specific notion of fairness their de-biasing method aims to achieve.  The authors start by listing applications like hiring and risk assessment in the criminal domain, but already these two examples operate under very different assumptions about the nature of bias, and the harms are also very different.  When referring to unfairness, are the authors referring to unequal error rates between groups (if so, what types of error specifically and why), or are they referring to disparities in outcomes (e.g., not enough women invited for job interviews)?  This has to be clarified upfront.  De-biasing is seen very often as a magic bullet, when publishing research results we must be very careful to define the scope of the problem and of the intervention, and to discuss the limitations.

2) The authors should acknowledge that being able to recover membership of an individual in a disadvantaged group may itself be highly problematic.  In fact, the doctrine of disparate treatment in the US disallows decision-makers in domains like credit and lending, hiring and employment, and others, to refer to a person's age, gender, disability status etc while making a decision that affects them.  The framework proposed by the authors would allow a malicious decision maker to recover a person's membership in a protected group.

The authors state: "It should be noted that identifying groups with positive or negative bias may not be same as identifying the sensitive attribute. The reason is that there may be multiple attributes that are highly correlated with the sensitive attribute. In such a situation, these correlated attributes can facilitate indirect discrimination even if the sensitive attribute is identified and removed. This is called the red-lining effect (Calders, 2010). Our model avoids this by directly identifying biased groups."   Yet, "directly identifying biased group membership", as denoted by the authors, can also be problematic, because it may allow a decision maker to conclude with high confidence that an individual is disabled (for example) even if they did not disclose their disability status in their job application, because others in that cluster stated that they are disabled.

---

> ### Author Response · Authors · 2021-08-18
> **Response to Ethics Reviewer Ag79**
>
> We welcome NeurIPS's decision to include ethics and societal implication evaluation to the review process. We propose to add the following discussion to the "Societal Impact Section" in the final version of the paper:
>
> "Fairness in machine learning is an active field of research. As it has the potential to affect the basic well being of our society, it should always be handled in consultation with field experts. This is especially important when one moves away from the theoretical setting and tries to apply fairness algorithms to real world data where the validity of technical assumptions cannot be easily verified. As with any algorithm in prior literature ([Calders et al., 2013; Agarwal et al., 2019; Fitzsimons et al.,2019, Berk et al.,2017, Chzhen et al.,2020]), one has to be extra cautious when interpreting the results of our method as well. In particular, we emphasize that our method does not characterize the nature of the recovered bias - good or bad. The nature of the bias should be determined by the field expert. We highly recommended to seek the field expert's input for interpreting the results in a real world setting. We advise to regress the discovered hidden attribute against some existing predictors provided by a field expert."
>
> We are open to discuss more suggestions from reviewers to add to this section.
>
> Below we answer specific questions from the reviewer.
>
> - The definition of bias follows the mathematical model of equation (1) where bias (additive) depends on a hidden attribute. This has been motivated verbally in second paragraph (Line 30). The first paragraph only gives an overview of many different settings for problems related to fairness. We aim to achieve the fairness notion of equal means (Calders, 2013) after debiasing. This is mentioned in Line 47-48.
>
> - Our method can only be used after the decision has been made by the decision maker. To that end, the decisions denoted by y in our formulation, check equation (1), are part of the samples we need. Even after this, as the reviewer notices, we cannot identify the sensitive attribute using our method. Any further speculations on identifying the sensitive attribute are just speculations and cannot be substantiated using our techniques.

---

### Decision · Program_Chairs · 2021-09-27

**Decision:**

Accept (Spotlight)

**Comment:**

In this work, the authors consider the problem of learning a fair sparse linear model when the sensitive attribute is binary and unknown. They establish that the solution of an invex optimization problem can recover both the correct support of the regression coefficients and the hidden sensitive labels. This problem and solution is of interest to the ML community as a variant of the generally important question of learning fair models that will be particularly apt in certain applications where the sensitive attribute is unknown.